# Diffusion-SS3D: Diffusion Model for Semi-supervised 3D Object Detection

**Cheng-Ju Ho**[1]     **Chen-Hsuan Tai**[1]     **Yen-Yu Lin**[1]     **Ming-Hsuan Yang**[2,3]     **Yi-Hsuan Tsai**[3]

[1]National Yang Ming Chiao Tung University     [2]University of California at Merced     [3]Google

## Abstract

Semi-supervised object detection is crucial for 3D scene understanding, efficiently addressing the limitation of acquiring large-scale 3D bounding box annotations. Existing methods typically employ a teacher-student framework with pseudo-labeling to leverage unlabeled point clouds. However, producing reliable pseudo-labels in a diverse 3D space still remains challenging. In this work, we propose Diffusion-SS3D, a new perspective of enhancing the quality of pseudo-labels via the diffusion model for semi-supervised 3D object detection. Specifically, we include noises to produce corrupted 3D object size and class label distributions, and then utilize the diffusion model as a denoising process to obtain bounding box outputs. Moreover, we integrate the diffusion model into the teacher-student framework, so that the denoised bounding boxes can be used to improve pseudo-label generation, as well as the entire semi-supervised learning process. We conduct experiments on the ScanNet and SUN RGB-D benchmark datasets to demonstrate that our approach achieves state-of-the-art performance against existing methods. We also present extensive analysis to understand how our diffusion model design affects performance in semi-supervised learning. The source code will be available at https://github.com/luluho1208/Diffusion-SS3D.

## 1 Introduction

3D object detection is a crucial computer vision task that predicts oriented bounding boxes and semantic classes of 3D objects for geometrical scene understanding, and thus it is essential to 3D applications such as autonomous driving, AR/VR, and robot navigation. Recent works [26, 30, 35, 39, 40, 41, 54, 59, 65] have made significant progress in the field of 3D object detection. However, most existing approaches heavily rely on labeled 3D point clouds, which requires substantial manual annotations. To overcome this limitation, semi-supervised learning (SSL) methods for 3D object detection have emerged, utilizing unlabeled 3D point clouds to supplement scarce labeled data and further improve detector performance.

Existing semi-supervised learning methods for 3D object detection [16, 53, 55, 60, 63] focus on generating pseudo-labels for 3D bounding boxes as supervisory signals on unlabeled data. However, different from the 2D cases where pseudo-labels are limited to the 2D image coordinates [7, 22, 25, 43, 44, 57, 64], the potential object locations in the 3D geometric space can be more diverse, causing the difficulty in generating high-quality pseudo-labels (i.e., pseudo bounding boxes) for 3D object detection. Consequently, an ensuing question arises: *Is there an effective approach for generating pseudo-labels that account for the vast 3D space in the realm of semi-supervised 3D object detection?* As shown in Figure 1(a), existing methods [16, 53, 63] usually adopt a teacher-student framework to provide pseudo-labels. In this case, bounding box candidates are solely based on model predictions, making it challenging to adjust pseudo bounding boxes, e.g., their sizes. In addition, pseudo-label quality may not be satisfactory when the model fails to output sufficient predictions (i.e., lower recall rate).

To mitigate this problem, we consider other sources to produce more reliable pseudo-labels for unlabeled data, and hence propose to denoise bounding boxes corrupted by random noise via the

37th Conference on Neural Information Processing Systems (NeurIPS 2023).

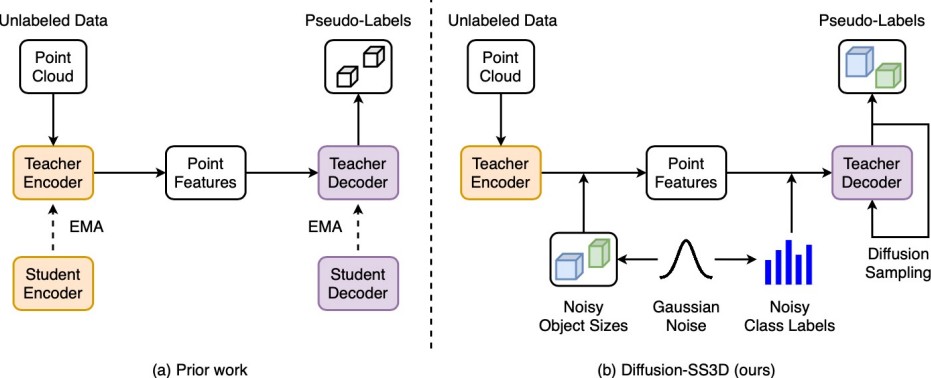

Figure 1: **Pseudo-label Generation.** (a) Prior works [16, 53, 63] apply a teacher-student framework where the point features serve as the only source for pseudo-label generation from the decoder output. (b) In our method, Diffusion-SS3D also adopts the teacher-student framework (the student model is omitted from the figure to save the space) and further integrates noisy object sizes and noisy label distributions in the denoising process (i.e., diffusion sampling). This enables more reliable pseudo-labels through the iterative refinement of denoising via the diffusion model.

diffusion model [13, 17, 45, 48]. There have been prior works using the diffusion model for 2D object detection [10] with purely random bounding boxes. However, applying the same process to 3D scenes is not trivial due to the vast 3D space, e.g., leading to a low recall rate of the ground-truth objects covered by purely random 3D object bounding boxes. To tackle this issue and generate plausible bounding box candidates, we include random noise in the 3D object size for representative points from the 3D model, in which each point indicates one potential object location. Therefore, our diffusion model starts from more accurate bounding box candidates with various object sizes, which may increase the chance (i.e., higher recall rate) of discovering better pseudo-labels for unlabeled data. Then, through the denoising process in our diffusion model, pseudo-labels are refined to be more reliable. Another challenge lies in less accurate class predictions for pseudo bounding boxes. To address it, we introduce a label denoising process, in which we add random noise to the class label distribution for each bounding box candidate, and then our diffusion model would denoise the class label to improve pseudo class label prediction.

We refer to our diffusion method for semi-supervised 3D object detection as *Diffusion-SS3D*, where we include random noise to 3D object size and class label distributions in the diffusion model to generate reliable pseudo bounding boxes (see Figure 1(b)). For SSL, we utilize the teacher-student framework [16, 53, 63] to incorporate labeled and unlabeled data into the training process, where the student model is trained based on (pseudo) ground truths and then update the teacher model with the exponential moving average (EMA) scheme [51] to produce pseudo-labels. To integrate the diffusion model into this framework, while training the student model, we regulate Gaussian noises to generate noisy object sizes and class labels as corrupted (pseudo) ground truths, followed by a decoder for predicting bounding box outputs. For the teacher model, we randomly generate object sizes and class label distributions, and apply diffusion sampling for denoising and producing pseudo-labels on unlabeled data. Then, during testing, we can utilize the diffusion sampler that denoises random object sizes and class labels to obtain bounding box predictions from the student model.

The proposed method, Diffusion-SS3D, explores a wider range of reasonable pseudo-label candidates for SSL in 3D object detection. We conduct experiments on the ScanNet [12] and SUN RGB-D [46] benchmark datasets to demonstrate that our approach achieves state-of-the-art performance. For example, our method improves the prior work by more than 6% in mAP@0.5 when using only the 1% labeled data on SUN RGB-D, and more than 5% when using 5% labeled data on ScanNet. This shows the benefit of adopting the proposed diffusion model to improve the pseudo-label quality. Moreover, we present extensive analysis to understand how our diffusion model design affects performance in SSL, e.g., ablation study of our diffusion model components, sampler steps, and signal-to-noise ratio (SNR). The main contributions of this work can be summarized as follows:

- We present Diffusion-SS3D, the first method to utilize the diffusion model for semi-supervised 3D object detection, treating the task as a denoising process for improving the quality of pseudo-labels.

- We propose to introduce the random noise to 3D object size and class label distributions for producing more plausible pseudo bounding boxes, by a means to integrate the diffusion model into the teacher-student framework for SSL.
- We present extensive experimental results to analyze the main factors in the diffusion model design that may affect 3D detector performance in SSL, as well as demonstrating state-of-the-art performance against existing methods.

## 2    Related Work

**Semi-supervised Learning.**    Semi-supervised learning (SSL) enhances models by incorporating unlabeled data during training. Typically SSL methods can be categorized into two groups: consistency regularization [5, 6, 21, 37, 56] and pseudo-labeling [22, 43, 51]. Among these approaches, the teacher-student framework [43, 51] is an effective self-training method, using two identical networks where the teacher model is guided by the student model via the exponential moving average mechanism. It also leverages asymmetric augmentations, using weak augmentation for the teacher model to ensure high-quality pseudo-label supervision and strong augmentation for the student model to enrich data variance for training.

**Semi-supervised Learning for Object Detection.**    For 2D object detection in SSL, similarly, existing methods adopt consistency regularization [19, 50] or teacher-student learning framework [7, 25, 43, 44, 57]. For 3D object detection, the primary paradigm in SSL [16, 29, 53, 63] employs the teacher-student framework with pseudo-labeling. For example, SESS [63] enforces consistency across different augmentations as regularization, while 3DIoUMatch [53] enhances the pseudo-label quality through confidence-based filtering with an IoU estimation module. Recently, OPA [16] introduces a trainable object-level augmentor to enrich the variance of foreground objects and improve data augmentation. DetMatch [29] and MVC-MonoDet [24] tackle the 3D SSL task via the multi-modal and multi-view settings respectively. However, most of these methods rely on the teacher model for pseudo-label generation, limiting the ability to identify bounding boxes that are not covered by model predictions. In this work, we propose to leverage the diffusion model to denoise from noisy object sizes and class labels as another source to provide high-quality pseudo-labels for bounding boxes.

**Diffusion Models for Visual Recognition.**    Diffusion models [17, 42, 47, 48] have attracted significant attention owing to their powerful denoising ability in generative models. Numerous fields have leveraged diffusion models, such as natural language processing [3, 18, 23, 38, 62], multimodal data generation [4, 32, 34, 36, 66], and temporal data modeling [1, 9, 20, 33, 52, 58]. Despite great success in synthesis tasks, diffusion models are less explored for visual recognition [2, 8, 10, 15, 27]. LRA [8] is introduced to mitigate the influence of noisy labels by utilizing neighbor consistency to create pseudo-clean labels for diffusion training. [27] utilizes diffusion models originally designed for image generation as a unified representation learner and demonstrates their effectiveness in extracting discriminative features for classification tasks. Moreover, DiffusionInst [15] formulates the instance segmentation task as a noise-to-filter problem, and utilizes the denoising capabilities of a diffusion model to generate better instance masks. DiffusionDet [10] casts object detection as a noise-to-box task, where high-quality object bounding boxes are produced by gradually denoising randomly generated proposals. In this work, we leverage the diffusion model in a different problem setting for semi-supervised learning, aiming to provide a new perspective of producing more reliable pseudo-labels for 3D object detection.

## 3    Methodology

### 3.1    Problem Definition and Preliminaries

**Semi-supervised 3D object detection.** 3D object detection aims to recognize and locate objects of interest in a 3D point cloud scene consisting of $N_p$ points $\mathbf{p} \in \mathbb{R}^{N_p \times 3}$, represented by their bounding boxes and semantic classes. Semi-supervised learning involves $N_l$ labeled scenes $\{\mathbf{p}_i^l, \mathbf{y}_i^l\}_{i=1}^{N_l}$ and $N_u$ unlabeled scenes $\{\mathbf{p}_i^u\}_{i=1}^{N_u}$, where typically $N_l << N_u$. Ground-truth annotations $\mathbf{y}_i^l$ contain $K$ objects of interest $\{\mathbf{b}_i^k, \mathbf{l}_i^k\}_{k=1}^{K}$ in $\mathbf{p}_i^l$, where $\mathbf{b}$ and $\mathbf{l}$ represent a set of bounding box parameters and semantic class labels respectively, with a total number of $N_{cls}$ classes. Specifically, we formulate the bounding box $\mathbf{b} = \{\mathbf{b}_c, \mathbf{b}_s, \mathbf{b}_o\}$, where $\mathbf{b}_c = \{c_x, c_y, c_z\}$ denotes the center coordinates, $\mathbf{b}_s = \{s_l, s_w, s_h\}$ represents the object size, and $\mathbf{b}_o$ corresponds to the bounding box orientation.

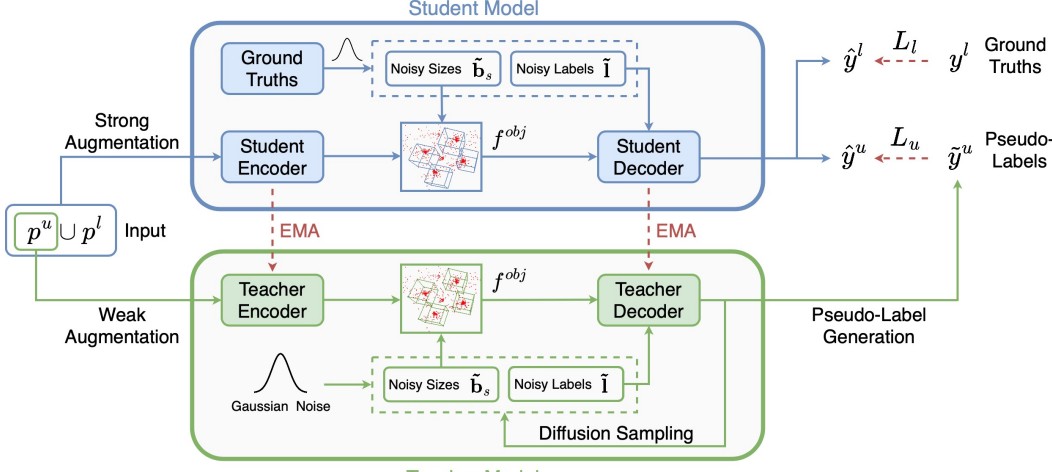

Figure 2: **Overview of Diffusion-SS3D.** Diffusion-SS3D employs the teacher-student learning framework combined with an asymmetric augmentation mechanism to incorporate both labeled and unlabeled data during training. In the student model, we establish a diffusion training process by introducing corrupted (pseudo) ground-truths by having the noisy object size $\tilde{\mathbf{b}}_s$ and noisy class label distributions $\tilde{\mathbf{l}}$. Then, based on the features $f^{obj}$ within boxes, the student decoder is trained to predict the bounding box output supervised by (pseudo) ground truths (see Figure 3 for details). For the teacher model updated by the student model via exponential moving average (EMA), we establish the diffusion inference process that denoises the randomly generated box sizes $\tilde{\mathbf{b}}_s$ and class label distributions $\tilde{\mathbf{l}}$ via iteratively applying diffusion sampling to output refined bounding box candidates. Following a filtering mechanism, the teacher decoder produces final pseudo-labels for unlabeled data.

**Diffusion model.** Diffusion models [13, 17, 45, 48] are generative models that simulate a diffusion process to represent data distributions. In the forward process, noise is progressively introduced to an initial data point, producing intermediate samples that converge to the target data distribution. The inference process estimates the posterior distribution of the initial data point from observed sample sequences by reversing the diffusion process. In [45], the diffusion forward process transforms an initial data sample $\mathbf{x}_0$ into a noisy sample $\mathbf{x}_t$ at time $t$ by adding noise governed by the noise variance schedule $\beta_t$ [17], with $\alpha_t := 1 - \beta_t$ and $\bar{\alpha}_t := \prod_{s=1}^{t} \alpha_s$. The forward process to generate a noisy sample $\mathbf{x}_t$ from $\mathbf{x}_0$ can be defined by

$$q\left(\mathbf{x}_t \mid \mathbf{x}_0\right) = \mathcal{N}\left(\mathbf{x}_t; \sqrt{\bar{\alpha}_t}\mathbf{x}_0, \left(1 - \bar{\alpha}_t\right)\mathbf{I}\right), \tag{1}$$

where $\mathcal{N}$ denotes a Gaussian distribution and $\mathbf{I}$ is an identity matrix. The model is trained to predict $\mathbf{x}_0$ from $\mathbf{x}_t$ by minimizing the $l_2$ loss between $\mathbf{x}_0$ and model outputs. During inference, $\mathbf{x}_0$ is obtained by gradually denoising from $\mathbf{x}_t$, i.e., $\mathbf{x}_t \to \mathbf{x}_{t-\Delta} \to \ldots \to \mathbf{x}_0$.

In this paper, we propose Diffusion-SS3D, a diffusion model for semi-supervised 3D object detection. We incorporate random noise for 3D object size and class label distributions in the diffusion model, and then reverse the diffusion process to generate reliable pseudo bounding boxes. Specifically, we define data samples as a set of bounding box parameters $\mathbf{x}_0 = \{\mathbf{b}_s, \mathbf{l}\}$, representing the 3D object size and one-hot semantic label respectively. Then our detector is trained to predict $\mathbf{x}_0$ from noisy bounding box parameters $\mathbf{x}_t$, conditioned on the features of the given point cloud $\mathbf{p}$. In SSL, we use this process to denoise bounding box candidates and generate final pseudo-labels for unlabeled data.

### 3.2 Algorithm Overview

Existing SSL methods for 3D object detection [16, 53, 63] utilize the teacher-student framework with asymmetric data augmentation to integrate labeled and unlabeled data in training. One critical factor is to generate pseudo-labels for unlabeled data, in which these approaches would solely rely on model outputs, which may restrict the models from generating better pseudo-labels (i.e., lower recall rate). Motivated by the challenge of creating high-quality pseudo-labels, we propose Diffusion-SS3D, a new perspective of producing reliable pseudo-labels via the diffusion model in the teacher-student framework. Figure 2 presents an overview of our method.

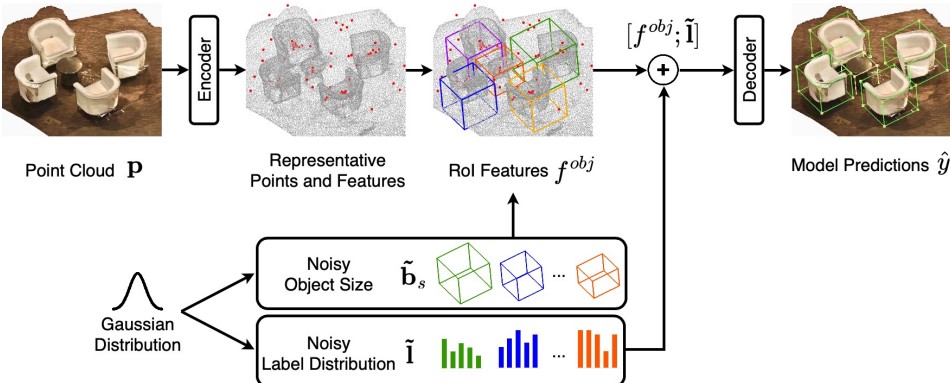

Figure 3: We illustrate the process of feeding RoI features with noisy object size and label distributions to the decoder before the diffusion sampling step. First, the input point cloud $\mathbf{p}$ is processed through the encoder, and the Farthest Point Sampling (FPS) is performed to obtain $N_b$ representative points $\{m_i\}_{i=1}^{N_b}$ (represented in red points). Concurrently, the noisy object size $\tilde{\mathbf{b}}_s$ and the noisy label distribution $\tilde{\mathbf{l}}$ are sampled from a Gaussian distribution. With representative points serving as centers of bounding boxes, RoI features $f^{obj}$ within each bounding box are extracted and then concatenated with noisy label distributions, together as the input to the decoder to predict bounding box output.

In the student model, the goal is to supervise model predictions, i.e., $\hat{\mathbf{y}}^l$ and $\hat{\mathbf{y}}^u$, for labeled data with ground truths $\mathbf{y}^l$ and unlabeled data with pseudo-labels $\tilde{\mathbf{y}}^u$, respectively. More importantly, we establish the training step for the diffusion process by generating the noisy data $\mathbf{x}_t = \{\tilde{\mathbf{b}}_s, \tilde{\mathbf{l}}\}_t$ from ground truths of the labeled data, forming noisy bounding box candidates, where $\tilde{\mathbf{b}}_s$ is the noisy object size and $\tilde{\mathbf{l}}$ is the noisy class label distribution. Then, we use the noisy bounding boxes, along with the corresponding features from the encoder, as the input to the decoder for making the predictions. Similarly, for the unlabeled data, the noisy bounding boxes can be generated from pseudo-labels. In each training iteration, the student model also updates the teacher model via an exponential moving average (EMA) mechanism for both the encoder and decoder.

In the teacher model, we aim to generate more reliable pseudo-labels for unlabeled data via the diffusion model. We start with randomly generated object sizes and class label distributions to form noisy bounding boxes, and then adopt the decoder to output the predictions similar to the process in the student model. Subsequently, we employ the diffusion sampling method [45] to iteratively denoise and produce more accurate object sizes and class labels for pseudo-label generation.

### 3.3 Diffusion Model Training

In this section, we introduce the components and procedure used to train the proposed Diffusion-SS3D framework, including the details of how we process point cloud features with generated noisy bounding boxes and class label distributions to train the diffusion model.

**Detector encoder.** The encoder takes a 3D point cloud scene $\mathbf{p} \in \mathbb{R}^{N_p \times 3}$ as input, executing only once during the diffusion training process to extract the feature representations $f \in \mathbb{R}^C$ for each point $m \in \mathbb{R}^3$ of $\mathbf{p}$ as a $C$-dimensional feature vector.

**Detector decoder.** Given a set of $N_b$ noisy object sizes $\tilde{\mathbf{b}}_s \in \mathbb{R}^{N_b \times 3}$ and noisy class labels $\tilde{\mathbf{l}} \in \mathbb{R}^{N_b \times N_{cls}}$, associated with their extracted RoI-features $f^{obj}$ within the object from the encoder, the decoder predicts bounding box regression values and classification results. Details of how the features are extracted are described in the following.

**RoI-feature extraction with noisy bounding boxes.** We aim to denoise from noisy bounding boxes to train the diffusion model. However, completely sampling random bounding boxes as the noisy data [10] may be too challenging for our case due to the vast 3D space. To increase the chance of producing plausible bounding box candidates with noise, we consider using representative point locations $\{m_i\}$ obtained from the 3D model, in which each point indicates a potential object location. Then, based on these points, we generate noisy bounding boxes by adding random noises to the 3D object size and class label distributions. The overall procedure is illustrated in Figure 3.

| **Algorithm 1:** Teacher Model | **Algorithm 2:** Student Model |
|---|---|

```
def pseudo_label(pc, steps, T):
  # Extract points and features
  pts, feats = Teacher.encoder(pc)

  # Noisy sizes and class labels
  sizes_t = normal(mean=0, std=1) # [N_b, 3]
  labels_t = normal(mean=0, std=1) # [N_b, N_cls]

  # Uniform sample step size
  times = reversed(linspace(-1, T, steps))
  time_pairs = list(zip(times[:-1], times[1:]))

  for t_cur, t_next in zip(time_pairs):
    # Generate noisy boxes
    boxes_t = concat(FPS(pts), sizes_t)

    # Predict pseudo labels
    pls = Teacher.decoder(feats, boxes_t,
                 lables_t, t_cur)

    # Estimate sizes_t and labels_t at t_next
    sizes_t, labels_t = ddim(sizes_t, labels_t,
                 pls, t_cur, t_next)

    # Box renewal
    sizes_t, labels_t = box_renewal(sizes_t,
                 labels_t)

  # Filtering
  pls = filter(pls)

  return pls
```
linespace:generate evenly spaced values
```
def object_prediction(pc, gts, pls):
  # Extract points and features
  pts, feats = Student.encoder(pc)

  # Pad to N_b bounding boxes
  sizes = prepare_size(gts, pls) # [N_b, 3]
  labels = prepare_label(gts, pls) # [N_b, N_cls]

  # Signal scaling
  sizes = (sizes * 2 - 1) * size_scale
  labels = (labels * 2 - 1) * label_scale

  # Corrupt ground truths
  t = randint(0, T) # time_step
  sizes_crpt = corrupt(sizes, t)
  labels_crpt = corrupt(labels, t)

  # Generate noisy boxes
  boxes_crpt = concat(FPS(pts), sizes_crpt)

  # Predict object candidates
  preds = Student.decoder(feats, boxes_crpt,
                 labels_crpt, t)

  # Supervise by (pseudo) ground truths
  loss = detector_loss(preds, gts, pls)
  Student = update(Student, loss)

  # Update teacher model
  Teacher = EMA(Teacher, Student)

  return preds
```
```
corrupt(x, t):sqrt(   alpha_cumprod(t)) * x +
           sqrt(1 - alpha_cumprod(t)) * noise
alpha_cumprod(t):∏ᵢ₌₁ᵗ α_i
```

Among all $M$ representative points generated by the encoder, we perform the Farthest Point Sampling (FPS) algorithm [31] to obtain a small set of $N_b$ sampled points $\{m_i\}_{i=1}^{N_b}$ that maximally cover this 3D scene. We then generate $N_b$ noisy bounding boxes by treating the sampled points $\{m_i\}_{i=1}^{N_b}$ as object centers, where each box has a noisy object size $\tilde{\mathbf{b}}_s$ and noisy class label distributions $\tilde{\mathbf{l}}$. From the encoded features $\{f_i\}$, we process features within each bounding box to form the RoI-features $\{f_i^{obj}\}_{i=1}^{N_b} \in \mathbb{R}^{N_b \times C}$. By concatenating the RoI-features and noisy class label distributions $\mathbf{f}_i^{obj} = [f_i^{obj}; \tilde{\mathbf{l}}] \in \mathbb{R}^{N_b \times (C+N_{cls})}$, the decoder takes $\mathbf{f}_i^{obj}$ as a noisy condition to predict the regression values for bounding boxes and classification results.

**Diffusion training procedure.** As illustrated in the upper part of Figure 2, the diffusion model is trained in the student model, where it involves adding Gaussian noise to (pseudo) ground truths and training the student decoder to reverse this process. In addition to corrupting (pseudo) ground truths, we generate random bounding boxes and class labels so that total $N_b$ noisy candidates are produced. Similar to [28], the magnitude of the noise is controlled by a cosine schedule that progressively lessens $\alpha_t$ in (1) along the iterative steps. Note that the coordinates of the ground truth box also need to be adjusted through signal scaling. This is critical as the signal-to-noise ratio (SNR) plays an important role in the performance of the diffusion model [11]. Then, the student model is trained by minimizing the loss between decoder's predictions and (pseudo) ground truths of labeled/unlabeled data. We present the training procedure in Algorithm 2.

## 3.4 Diffusion Inference for Pseudo-Labels

As illustrated in the lower part of Figure 2, we perform the diffusion inference step in the teacher model to denoise noisy bounding box candidates for unlabeled data. To be specific, we randomly generate the initial noisy data $\mathbf{x}_t = \{\tilde{\mathbf{b}}_s, \tilde{\mathbf{l}}\}_t$ from a Gaussian distribution and extract $N_b$ candidate features $\mathbf{f}^{obj} = [f^{obj}; \tilde{\mathbf{l}}]$ as those in the diffusion training step. Then, the decoder would output model predictions based on $\mathbf{f}^{obj}$, followed by the DDIM sampling [17] to iteratively denoise $\mathbf{x}_t$ and achieve better states, e.g., closer to $\mathbf{x}_0$. Finally, we utilize the denoised model predictions and generate pseudo-labels for unlabeled data. Algorithm 1 shows the overall inference procedure.

**Bounding box renewal.** During the iterative DDIM sampling step, having a filtering scheme can early reject some low-quality candidates that are unlikely to be the final pseudo-labels. Therefore, we apply a bounding box renewal mechanism at each sampling step similar to [10] and replace them by sampling new box sizes with class label distributions from a Gaussian distribution. Specifically, we adopt the filtering scheme originally used in SSL for 3D object detection [53], where the confidence score of each candidate is estimated based on the three metrics: objectness, classification confidence, and intersection-over-union (IoU) scores.

### 3.5 Overall Procedure in SSL

**Training objectives for SSL.** Our training procedure consists of two phases: pre-training and SSL training. We begin by training the proposed Diffusion-SS3D through supervised learning using labeled data $\{\mathbf{p}_i^l, \mathbf{y}_i^l\}_{i=1}^{N_l}$ to pre-train our detector. Next, we initialize the teacher an student models with the same detector weights, and then jointly train the detector using both labeled data $\{\mathbf{p}_i^l, \mathbf{y}_i^l\}_{i=1}^{N_l}$ and unlabeled data $\{\mathbf{p}_i^u\}_{i=1}^{N_u}$, where each training batch includes a mixture of labeled and unlabeled scenes. The student model receives both labeled and unlabeled data after strong augmentation, while the unlabeled data with weak augmentation are fed into the teacher model to produce pseudo-labels $\tilde{\mathbf{y}}^u$ through iterative diffusion sampling. The overall loss function $\mathcal{L}$ for the student model is defined as $\mathcal{L} = \mathcal{L}_l(\mathbf{p}^l, \mathbf{y}^l) + \lambda \mathcal{L}_u(\mathbf{p}^u, \tilde{\mathbf{y}}^u)$, where the $\mathcal{L}_l$ and $\mathcal{L}_u$ are the detection loss functions used in [53] for bounding box regression and classification. $\lambda$, set as 2 following [53], is the loss weight to balance the importance of labeled and unlabeled data.

**Model evaluation.** We utilize the student model to perform the testing stage. Similar to the inference step in Section 3.4, randomly generated bounding box candidates are denoised through the DDIM sampling and bounding box renewal steps to produce final 3D object detection results.

## 4 Experiments

**Datasets.** We evaluate our method on two benchmarks, including the ScanNet [12] and SUN RGB-D [46] datasets, with the evaluation settings adopted in the prior semi-supervised 3D object detection works [16, 53, 63]. ScanNet [12], a widely used 3D indoor scene benchmark dataset, consists of 1,201 training and 312 validation indoor scenes reconstructed from 2.5 million high-quality RGB-D images. Following the previous work [16, 53, 63], we primarily focus on the 18 semantic classes. SUN RGB-D [46], another 3D benchmark dataset, comprises 5,285 training and 5,050 validation scenes, where we evaluate our model on the 10 object classes.

**Evaluation metrics.** We split both benchmarks into labeled and unlabeled data for SSL, using labeled data ratios of 5%, 10%, and 20% for ScanNet and 1%, 5%, and 10% for SUN RGB-D. We evaluate our results using mean average precision (mAP) and report mAP@0.25 (mAP with the 3D IoU threshold of 0.25) and mAP@0.5 scores. We run the experiments under three random data splits and report our averaged performance and the standard deviation.

**Implementation details.** In this work, we employ PointNet++ [31] as the encoder and IoU-aware VoteNet [53] as the decoder. In the pre-training phase, we only use labeled data with a batch size of 4 to train the diffusion model. The model is trained for 900 epochs with an initial learning rate of 0.005. Like [16, 53], the learning rate then decays at the 400th, 600th, and 800th epochs with a factor of 0.1. In the phase of semi-supervised learning, a batch is composed of 4 labeled and 8 unlabeled data. The pre-trained model is used for initializing both the teacher and student models. The student model is trained for 1,000 epochs using the AdamW optimizer, with an initial learning rate of 0.005. Like [16, 53], the learning rate decays at the 400th, 600th, 800th, and 900th epochs with factors of 0.3, 0.3, 0.1, and 0.1, respectively.

For the diffusion process, we set the maximum timesteps to 1000 and the number of proposal boxes $N_b$ to 128. We note that during random sampling from the Gaussian distribution, objects are typically present in a small space of a 3D scene. Therefore, different from DiffusionDet [10], which sets the mean of the sampling sizes to 0.5, we use a smaller value of 0.25. For noisy labels, the mean of random sampling is adjusted to the reciprocal of the class number. Unless otherwise specified, we perform two DDIM sampling [45] steps in the teacher model to generate pseudo-labels, as well as during evaluation to produce final results. For fair comparisons, we follow 3DIoUMatch [53] to employ a pseudo-label filtering scheme as a post-processing step on top of our generated pseudo-labels via the diffusion process.

Table 1: Results on the ScanNet val set with 5%, 10%, 20%, and 100% labeled data.

| Model | 5% | | 10% | | 20% | | 100% | |
|---|---|---|---|---|---|---|---|---|
| | mAP @ 0.25 | mAP @ 0.5 | mAP @ 0.25 | mAP @ 0.5 | mAP @ 0.25 | mAP @ 0.5 | mAP @ 0.25 | mAP @ 0.5 |
| VoteNet [30] | 27.9 ± 0.5 | 10.8 ± 0.6 | 36.9 ± 1.6 | 18.2 ± 1.0 | 46.9 ± 1.9 | 27.5 ± 1.2 | 57.8 | 36.0 |
| SESS [63] | 32.0 ± 0.7 | 14.4 ± 0.7 | 39.5 ± 1.8 | 19.8 ± 1.3 | 49.6 ± 1.1 | 29.0 ± 1.0 | 61.3 | 38.8 |
| 3DIoUMatch [53] | 40.0 ± 0.9 | 22.5 ± 0.5 | 47.2 ± 0.4 | 28.3 ± 1.5 | 52.8 ± 1.2 | 35.2 ± 1.1 | 62.9 | 42.1 |
| Diffusion-SS3D | **43.5 ± 0.2** | **27.9 ± 0.3** | **50.3 ± 1.4** | **33.1 ± 1.5** | **55.6 ± 1.7** | **36.9 ± 1.4** | **64.1** | **43.2** |
| Gain (mAP) | 3.5↑ | 5.4↑ | 3.1↑ | 4.8↑ | 2.8↑ | 1.7↑ | 1.2↑ | 1.1↑ |

Table 2: Results on the SUN RGB-D val set with 1%, 5% 10%, and 20% labeled data.

| Model | 1% | | 5% | | 10% | | 20% | |
|---|---|---|---|---|---|---|---|---|
| | mAP @ 0.25 | mAP @ 0.5 | mAP @ 0.25 | mAP @ 0.5 | mAP @ 0.25 | mAP @ 0.5 | mAP @ 0.25 | mAP @ 0.5 |
| VoteNet [30] | 18.3 ± 1.2 | 4.4 ± 0.4 | 29.9 ± 1.5 | 10.5 ± 0.5 | 38.9 ± 0.8 | 17.2 ± 1.3 | 45.7 ± 0.6 | 22.5 ± 0.8 |
| SESS [63] | 20.1 ± 0.2 | 5.8 ± 0.3 | 34.2 ± 2.0 | 13.1 ± 1.0 | 42.1 ± 1.1 | 20.9 ± 0.3 | 47.1 ± 0.7 | 24.5 ± 1.2 |
| 3DIoUMatch [53] | 21.9 ± 1.4 | 8.0 ± 1.5 | 39.0 ± 1.9 | 21.1 ± 1.7 | 45.5 ± 1.5 | 28.8 ± 0.7 | 49.7 ± 0.4 | 30.9 ± 0.2 |
| Diffusion-SS3D | **30.9 ± 1.0** | **14.7 ± 1.2** | **43.9 ± 0.6** | **24.9 ± 0.3** | **49.1 ± 0.5** | **30.4 ± 0.7** | **51.4 ± 0.8** | **32.4 ± 0.6** |
| Gain (mAP) | 9.0↑ | 6.7↑ | 4.9↑ | 3.8↑ | 3.6↑ | 1.6↑ | 1.7↑ | 1.5↑ |

## 4.1 Main Results

Tables 1 and 2 show the results of Diffusion-SS3D on the ScanNet and SUN RGB-D datasets with different amounts of labeled data. Overall, our method performs favorably against state-of-the-art approaches, including VoteNet [30], SESS [63], and 3DIoUMatch [53]. In particular, comparing to the main baseline, 3DIoUMatch, Diffusion-SS3D achieves consistent performance improvement across all settings, especially when fewer labeled data are available. For example, for challenging scenarios like 5% ScanNet and 1% SUN RGB-D settings, our method improves the mAP@0.5 metric by more than 5% in both settings. For higher labeled data ratios or in the fully-supervised setting (i.e., 100%), Diffusion-SS3D demonstrates a consistent 1% improvement in both mAP metrics. The results demonstrate the effectiveness of using the diffusion model to produce more reliable pseudo-labels in SSL and in the fully-supervised scenario (even though it is not our main focus).

In addition, we apply our diffusion model to a data augmentation method, OPA [16], which enhances 3D object detection in SSL via learning a point augmentor in the teacher-student framework. Since OPA works on data augmentation on the data input side, we integrate it into our Diffusion-SS3D framework and conduct experiments to show the complementary benefit. Specifically, when using 5% labeled data in ScanNet, our method achieves a mAP@0.25 of 44.6% and a mAP@0.5 of 28.1%, which are +2.7% and +3.1% better than the OPA approach, respectively. This demonstrates the usefulness of our approach combined with other data augmentation techniques. More details and results are provided in the supplementary materials.

## 4.2 Ablation Study and Analysis

**Diffusion on object sizes and labels.** In Table 3, we delve into the impact of our proposed diffusion model by focusing on two primary aspects: diffusion on object sizes and class label distributions. With the denoising process in object sizes, i.e., ID(2), the performance is improved significantly compared to the baseline without the diffusion model in ID(1). Moreover, adding label denoising further enhances the quality of pseudo-labels and thus achieves the best performance in ID(4). Interestingly, we find that the effectiveness of size denoising is slightly better than label denoising, i.e., comparing ID(2) and (3). Here, we consider this a task-specific behavior, where the parameters to be denoised may have different influences on various perception tasks.

**DDIM and box renewal.** As described in Section 3.4, DDIM sampling [45] and a box renewal mechanism are used to infer pseudo-labels in the teacher model. In Table 4, we validate the importance of these two components. First, comparing ID(1) and (2), DDIM sampling contributes via denoising object sizes and class label distributions to obtain better qualities of bounding box candidates for pseudo-label generation. Then, comparing ID(1) and (3), it shows that box renewal also has the denoising capability. We note that the effectiveness of box renewal is also enhanced through the diffusion training process, where it utilizes the trained diffusion decoder to integrate updated bounding box features and label distributions. Finally, the best performance is achieved in ID(4) with both DDIM and box renewal, validating our diffusion model's capability in producing reliable pseudo-labels.

Table 3: Ablation study on the effect of box size and class label distribution in the diffusion process.

| ID | Diffusion on Sizes | Diffusion on Labels | ScanNet 5% | | SUNRGBD 1% | |
|----|:---:|:---:|:---:|:---:|:---:|:---:|
| | | | mAP @ 0.25 | mAP @ 0.5 | mAP @ 0.25 | mAP @ 0.5 |
| (1) | | | 40.5 ± 1.2 | 22.8 ± 0.8 | 21.9 ± 1.4 | 8.0 ± 1.5 |
| (2) | ✓ | | 43.1 ± 0.3 | 26.3 ± 0.2 | 30.5 ± 0.3 | 13.8 ± 0.7 |
| (3) | | ✓ | 42.6 ± 0.6 | 26.0 ± 0.5 | 30.3 ± 0.4 | 13.3 ± 0.2 |
| (4) | ✓ | ✓ | **43.5** ± 0.2 | **27.9** ± 0.3 | **30.9** ± 1.0 | **14.7** ± 1.2 |

Table 4: Ablation study on the effect of DDIM and box renewal.

| ID | DDIM | Box Renewal | ScanNet 5% | | SUNRGBD 1% | |
|----|:---:|:---:|:---:|:---:|:---:|:---:|
| | | | mAP @ 0.25 | mAP @ 0.5 | mAP @ 0.25 | mAP @ 0.5 |
| (1) | | | 40.5 ± 1.2 | 22.8 ± 0.8 | 21.9 ± 1.4 | 8.0 ± 1.5 |
| (2) | ✓ | | 42.8 ± 0.5 | 26.6 ± 0.9 | 30.2 ± 0.7 | 13.9 ± 1.3 |
| (3) | | ✓ | 42.3 ± 0.4 | 26.8 ± 0.3 | 29.5 ± 0.9 | 12.8 ± 0.6 |
| (4) | ✓ | ✓ | **43.5** ± 0.2 | **27.9** ± 0.3 | **30.9** ± 1.0 | **14.7** ± 1.2 |

Table 5: Ablation study on various diffusion sampling steps.

| ID | Num. DDIM | ScanNet 5% | | SUNRGBD 1% | |
|----|:---:|:---:|:---:|:---:|:---:|
| | | mAP @ 0.25 | mAP @ 0.5 | mAP @ 0.25 | mAP @ 0.5 |
| (1) | 1 | 42.6 ± 0.5 | 26.2 ± 0.7 | 30.3 ± 0.7 | 12.7 ± 0.2 |
| (2) | 2 | 43.5 ± 0.2 | **27.9** ± 0.3 | **30.9** ± 1.0 | **14.7** ± 1.2 |
| (3) | 4 | **44.0** ± 1.1 | 27.5 ± 1.0 | 28.0 ± 1.1 | 12.9 ± 0.3 |

Table 6: Ablation study on various scaling factors for SNR.

| ID | Scaling Factor | ScanNet 5% | | SUNRGBD 1% | |
|----|:---:|:---:|:---:|:---:|:---:|
| | | mAP @ 0.25 | mAP @ 0.5 | mAP @ 0.25 | mAP @ 0.5 |
| (1) | 1.0 | 43.2 ± 0.3 | 26.6 ± 0.9 | 30.5 ± 0.8 | 13.2 ± 0.5 |
| (2) | 2.0 | 43.5 ± 0.2 | 26.7 ± 0.2 | **31.3** ± 1.1 | 14.2 ± 1.4 |
| (3) | 4.0 | **43.5** ± 0.2 | **27.9** ± 0.3 | 30.9 ± 1.0 | **14.7** ± 1.2 |

**Number of DDIM steps.** We study how the diffusion sampling steps, i.e., DDIM [45], affect the performance. In general, having more sampling steps may better remove noises and obtain outputs closer to ground truths. However, since we address SSL, the denoising steps may have a different influence on unlabeled data because we do not have its ground truth for training the diffusion model (e.g., a distribution shift from labeled data). In Table 5, we find that having two DDIM steps improves the denoising process, but having four steps may slightly degrade the performance, which is an interestingly finding for SSL. Hence we adopt two sampling steps in all the experiments. Note that, the observed sensitivity of DDIM steps to different datasets is partially attributed to limited labeled data, which may make diffusion sampling steps more sensitive when inferring unlabeled data distribution.

**Signal-to-noise ratio.** The signal scaling factor is a critical parameter that can affect the signal-to-noise ratio (SNR) during the diffusion process. Table 6 explores the impact of various scaling factors. Overall, our method performs stably with different values, and we choose 4.0 as our final setting in all the experiments. It is also worth noting that, in our context like class label distributions, e.g., the 18 classes in ScanNet, there are much fewer parameters compared to more complicated diffusion tasks, e.g., image generation. Therefore, we find that having a higher SNR can be slightly beneficial and make training more stable, as it reduces the difficulty of the denoising process.

**Pseudo-label quality.** To assess the enhanced quality of pseudo-labels generated by Diffusion-SS3D, we evaluate metrics on unlabeled training data during model training, via the teacher model that generates pseudo-labels. In Table 7, we demonstrate that overall the pseudo-label quality of Diffusion-SS3D outperforms 3DIoUMatch [53] by more than 8% improvement in mAP and recall rate. Notably, our diffusion model achieves better quality in earlier epochs and maintains stability throughout the entire training process. It is important to highlight that the semi-supervised performance continues to improve as the model learns from both labeled and unlabeled data over extended training periods. These results show the ability of the diffusion model to consistently generate high-quality pseudo-labels. For visual comparisons of the generated pseudo-labels, please refer to the supplementary materials.

Table 7: Results regarding the quality of pseudo-labels generated for unlabeled training data by the teacher model during the training process.

| Model | Metric | ScanNet 5% | | | | |
|---|---|---|---|---|---|---|
| | | Epoch 100 | Epoch 200 | Epoch 400 | Epoch 800 | Epoch 1000 |
| 3DIoUMatch | mAP @ 0.5 | 14.08 | 17.85 | 21.73 | 22.17 | 22.42 |
| | Recall @ 0.5 | **27.86** | **31.49** | **35.24** | **36.61** | **35.25** |
| Diffusion-SS3D | mAP @ 0.5 | 29.98 | 30.09 | 30.86 | 31.01 | 30.93 |
| | Recall @ 0.5 | **43.73** | **44.14** | **45.06** | **44.72** | **44.17** |

Table 8: Ablation study on the effect of the diffusion component and various sampling strategies.

| ID | Diffusion | Sampling Stragety | ScanNet 5% | |
|---|---|---|---|---|
| | | | mAP @ 0.25 | mAP @ 0.5 |
| (1) | | Random | 40.2 ± 1.5 | 22.3 ± 1.1 |
| (2) | | Farthest Point | 40.5 ± 1.2 | 22.8 ± 0.8 |
| (3) | ✓ | Random | 43.1 ± 0.6 | 27.4 ± 0.6 |
| (4) | ✓ | Farthest Point | **43.5 ± 0.2** | **27.9 ± 0.3** |

**Farthest point sampling.** In our approach, point sub-sampling is employed to narrow the search space and limit freedom in generating noisy boxes, facilitating the diffusion learning process. We explore different sampling strategies, including random point sub-sampling and farthest point sampling (FPS), to select representative centers, as shown in Table 8. Comparing ID(1) and ID(2) reveals similar performance when applying random sampling and FPS to the 3DIoUMatch [53] model without the diffusion component. Similarly, ID(3) and ID(4) show comparable results for random sampling and FPS with our approach. Notably, our diffusion model, utilizing both FPS and random sampling, outperforms the 3DIoUMatch [53] baseline. In practice, we apply FPS following the methods of VoteNet [30] and 3DIoUMatch [53].

## 4.3 More Discussions

**Object orientation.** In this paper, we focus on denoising object size and class labels, while considering orientations can be another potential direction for our diffusion model. However, incorporating orientation data is challenging due to the discrepancy in orientation quality between ground truths and predictions, as noted in [61]. For example, ScanNet [12] does not provide orientations, i.e., all objects are assigned with orientations as 0. Similarly, in SUN RGB-D [46], orientation data varies inconsistently across scenes, complicating the denoising process. Exploring diffusion models capable of handling diverse data types, including noisy ground truths and orientation information, remains an intriguing topic for future research.

**Outdoor datasets.** Diffusion-SS3D excels on indoor benchmark datasets like ScanNet [12] and SUN RGB-D [46] against existing state-of-the-art methods [16, 53]. Although our approach is not restricted to specific datasets (i.e., integrating diffusion model within a general teacher-student framework), exploring outdoor benchmark datasets like KITTI [14] and Waymo Open Dataset [49] may require hyperparameter changes since some object properties (e.g., location, density) in outdoor would be different from the indoor scenario.

**Societal impact.** Since the diffusion models require more computational powers for training and inference (runtime in frame-per-second is presented in the supplementary materials), optimizing efficiency is crucial to reduce the environmental footprint for real-time applications and large-scale deployments.

## 5 Conclusions

In this paper, we address the challenge of generating high-quality pseudo-labels for 3D object detection in semi-supervised learning. We propose Diffusion-SS3D, a new perspective of employing the diffusion model to produce reliable pseudo-labels. via adding random noise to 3D object sizes and class label distributions, and subsequently reversing the diffusion process We then incorporate the diffusion model into a teacher-student framework, which facilitates the learning process with denoised bounding box candidates towards better pseudo-labels. We evaluate the effectiveness of Diffusion-SS3D on the ScanNet and SUN RGB-D benchmark datasets and demonstrate the state-of-the-art performance, with extensive analysis of how the diffusion model affects SSL performance. Our method explores a broader range of reasonable pseudo-label candidates for SSL in 3D object detection, showing the potential of the diffusion model in this domain.

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
