# Diffusion-SS3D: Diffusion Model for Semi-supervised 3D Object Detection
# – Supplementary Material –

## A   More Implementation Details

The proposed Diffusion-SS3D utilizes a teacher-student framework for 3D object detection in the setting of semi-supervised learning (SSL) and leverages the PointNet++ [3] as the encoder and the IoU-aware VoteNet [4] as the diffusion decoder. This section provides more details about the components in our implementation, including the encoder, decoder, diffusion initialization, SSL loss functions, and pseudo-label generation.

**Encoder.** In the encoder, we process an input point cloud using PointNet++ [3] to extract 1024 high-level representative seed points, denoted as $m = \{m_i \in \mathbb{R}^3\}_{i=1}^{1024}$, and their corresponding $C$-dimensional features, denoted as $f = \{f_i \in \mathbb{R}^C\}_{i=1}^{1024}$. In VoteNet, all seed points $m$ contribute to voting and generate their corresponding vote points, denoted as $v = \{v_i \in \mathbb{R}^3\}_{i=1}^{1024}$, and their vote features $f^v = \{f_i^v \in \mathbb{R}^C\}_{i=1}^{1024}$, where each vote point represents a potential object center. Subsequently, we perform Farthest Point Sampling (FPS) on the seed points to obtain $N_b$ sampled seed points to maximally cover the 3D scene. Without loss of generality, the sampled points are denoted by $\{m_i\}_{i=1}^{N_b}$. The corresponding vote points $\{v_i\}_{i=1}^{N_b}$ of these seed points are used as the $N_b$ random box centers, along with their features $f^v$, used in the next step.

**Preparing noisy boxes.** To start the diffusion training process with corrupted ground truths, a ground-truth box center $\mathbf{b}_c$ is first matched to the nearest seed point among $\{m_i\}_{i=1}^{N_b}$. Then its corresponding vote point, $v_i$, is treated as the proposal box center. Next, the noisy object size and noisy class label distributions are generated by adding Gaussian noise to the ground truth, and are used for this proposal box as the corrupted ground truth.

**Decoder.** In the decoding phase, for each vote point $v$ used as the object center, we extract its RoI-features $f^{obj}$ by employing a 4-layer MLP with max pooling to aggregate all the features $f^v$ within its noisy bounding box. Then, the RoI-features $f^{obj}$ are concatenated with the noisy label distributions $\tilde{\mathbf{l}}$ to form the features $\mathbf{f}^{obj} = [f^{obj}; \tilde{\mathbf{l}}] \in \mathbb{R}^{C+N_{cls}}$ (see Figure 3 of the manuscript), which serve as the input to the decoder for bounding box regression and classification.

**Loss functions for object detection.** To train Diffusion-SS3D under the SSL setting, we supervise the student model using ground-truths $\mathbf{y}^l$ for labeled data $\mathbf{p}^l$ and pseudo-labels $\tilde{\mathbf{y}}^u$ obtained from the teacher model for unlabeled data $\mathbf{p}^u$, respectively. The overall loss function $\mathcal{L}$ for the student model is formulated as $\mathcal{L} = \mathcal{L}_l(\mathbf{p}^l, \mathbf{y}^l) + \lambda \mathcal{L}_u(\mathbf{p}^u, \tilde{\mathbf{y}}^u)$, where $\mathcal{L}_l$ and $\mathcal{L}_u$ represent the detection loss functions for bounding box regression and classification, applied to the labeled and unlabeled data, respectively. In our experiments, we follow [4] and set $\lambda$ to 2 for balancing the importance of labeled and unlabeled data. Both $\mathcal{L}_l$ and $\mathcal{L}_u$ employ the IoU-aware VoteNet loss [4], which contains two components, including the original VoteNet losses [2], denoted as $L_{vote}$, and the 3D IoU loss [4], denoted as $L_{IoU}$. The main objective $L_{vote}$ includes various losses such as vote coordinate regression, objectness score binary classification, box center regression, bin classification, residual regression for heading angle, scale regression, and category classification.

**Pseudo-label generation.** During the pseudo-label generation process in the teacher model with diffusion sampling, we use the filtering scheme [4] based on objectness, classification confidence, and intersection-over-union (IoU) scores. Following [4], we set the threshold to 0.9 for the objectness and classification confidence scores, and the threshold to 0.25 for the IoU score.

Table 1: Per-class AP@0.25 (top group) and AP@0.5 (bottom group) on the ScanNet val set with 5% labeled data.

| | cabin | bed | chair | sofa | table | door | wind | bkshf | pic | cntr | desk | curt | refrig | showr | toilet | sink | bath | ofurn |
|---|---|---|---|---|---|---|---|---|---|---|---|---|---|---|---|---|---|---|
| VoteNet [2] | 10.4 | 65.3 | 73.3 | 67.9 | 19.7 | 11.0 | 5.6 | 11.6 | **2.4** | 17.2 | 39.0 | 15.3 | 8.5 | 10.4 | 68.8 | 16.8 | 54.9 | 7.5 |
| SESS [5] | 13.1 | 75.8 | 69.8 | 70.7 | 35.4 | 10.4 | 10.5 | 7.5 | **2.4** | 10.2 | 39.6 | 20.9 | 13.9 | 20.4 | 74.7 | **22.1** | 62.2 | 7.4 |
| 3DIoUMatch [4] | 20.8 | 77.4 | 78.1 | 75.6 | 43.5 | 14.6 | 18.9 | 16.9 | 1.5 | 29.4 | 44.4 | 26.0 | 27.5 | 36.3 | 80.4 | 18.3 | **82.5** | 13.0 |
| Ours | **21.8** | **83.7** | **81.9** | **79.2** | **47.1** | **26.5** | **21.6** | **32.8** | 1.2 | **52.7** | **47.0** | **31.5** | **28.0** | **37.2** | **90.0** | 20.0 | 59.7 | **19.1** |
| VoteNet [2] | 0.9 | 52.1 | 36.1 | 39.0 | 4.2 | 1.0 | 0.1 | 4.1 | 0.0 | 0.4 | 10.9 | 1.6 | 2.8 | 0.1 | 30.3 | 2.0 | 38.5 | 0.6 |
| SESS [5] | 3.6 | 63.6 | 40.5 | 51.8 | 14.8 | 1.9 | 0.6 | 5.3 | 0.2 | 0.2 | 11.9 | 7.6 | 4.1 | 0.6 | 52.7 | 7.4 | 45.4 | 1.6 |
| 3DIoUMatch [4] | **6.4** | 64.0 | 52.9 | 57.7 | 22.5 | 2.4 | 3.6 | 3.3 | **0.3** | 7.7 | 21.1 | 12.4 | 12.7 | 4.9 | 70.4 | **10.3** | **59.0** | 3.9 |
| Ours | 4.5 | **65.1** | **59.3** | **67.3** | **32.8** | **5.7** | **4.5** | **22.7** | **0.3** | **28.7** | **30.5** | **12.5** | **20.5** | **6.3** | **72.9** | 8.1 | 47.6 | **6.8** |

Table 2: Per-class AP@0.25 (top group) and AP@0.5 (bottom group) on the SUN RGB-D val set with 1% labeled data.

| | bathtub | bed | bookshelf | chair | desk | dresser | nightstand | sofa | table | toilet |
|---|---|---|---|---|---|---|---|---|---|---|
| VoteNet [2] | 49.0 | 15.9 | 19.0 | 34.6 | 20.8 | 4.8 | 1.2 | 4.6 | 1.8 | 30.6 |
| SESS [5] | 56.4 | 18.1 | 28.8 | 42.6 | 26.4 | **5.7** | 4.3 | 11.1 | 1.6 | 37.0 |
| 3DIoUMatch [4] | 60.3 | 22.2 | 30.7 | 45.3 | 25.7 | 3.6 | 2.5 | 15.6 | 1.2 | 38.0 |
| Ours | **67.5** | **30.6** | **40.5** | **52.8** | **56.4** | 3.4 | **4.8** | **16.0** | **2.9** | **47.3** |
| VoteNet [2] | 25.1 | 1.3 | 3.2 | 10.7 | 4.5 | 0.1 | 0.1 | 0.6 | 0.3 | 3.1 |
| SESS [5] | 27.0 | 2.1 | 8.2 | 13.2 | 5.4 | 0.4 | 0.9 | 1.5 | 0.1 | 4.7 |
| 3DIoUMatch [4] | 27.8 | 6.1 | 11.9 | 19.0 | 8.2 | 0.7 | 1.2 | 3.9 | 0.3 | 13.5 |
| Ours | **37.2** | **10.3** | **22.9** | **28.8** | **25.0** | **1.2** | **1.7** | **5.3** | **0.7** | **16.9** |

Table 3: Results on the ScanNet val set with 5%, 10%, and 20% labeled data using our method with OPA as the data augmentation technique.

| Model | 5% | | 10% | | 20% | |
|---|---|---|---|---|---|---|
| | mAP@0.25 | mAP@0.5 | mAP@0.25 | mAP@0.5 | mAP@0.25 | mAP@0.5 |
| OPA [1] | 41.9 ± 1.5 | 25.0 ± 0.4 | 50.5 ± 0.2 | 32.7 ± 1.0 | 54.7 ± 0.3 | 36.8 ± 0.8 |
| OPA + ours | **44.1** ± 0.6 | **27.4** ± 0.6 | **52.3** ± 0.6 | **36.5** ± 0.7 | **55.6** ± 0.5 | **38.8** ± 0.6 |
| Gain (mAP) | 2.2↑ | 2.4↑ | 1.8↑ | 3.8↑ | 0.9↑ | 2.0↑ |

# B  More Experimental Results

## B.1  Per-class AP Evaluation

In Table 1 and Table 2, we present the average precision per class for both the ScanNet validation set with 5% labeled data and the SUN RGB-D validation set with 1% labeled data, respectively. Overall, Diffusion-SS3D exhibits consistent performance improvement compared to existing state-of-the-art methods, underlining the benefit of our diffusion model, which can effectively denoise from noisy sizes and noisy label distributions, and thus yield more reliable pseudo-labels.

## B.2  Data Augmentation

In this section, we apply our diffusion model to a data augmentation method, OPA [1], which enhances 3D object detection in SSL via learning a point augmentor in the teacher-student framework. Since OPA focuses on data augmentation for the input data, we seamlessly integrate it into our Diffusion-SS3D framework and conduct experiments to demonstrate the complementary benefits. To begin, we train an augmentor using labeled data, following OPA's official implementation. We then employ this augmentor to generate augmented data, which serves as the input to our framework. Note that we maintain all other diffusion components identical to what is described in the manuscript. As shown in Table 3, incorporating OPA [1] as an augmentation technique in our diffusion model processing yields state-of-the-art results. For example, when utilizing 5% labeled data in ScanNet, our method achieves a mAP@0.25 of 44.6% and a mAP@0.5 of 28.1%, which are +2.7% and +3.1% better than the OPA [1] approach, respectively. Overall, we show that our diffusion model is complementary to additional data augmentation and performs favorably against OPA [1]. This demonstrates the effectiveness of our method when combined with other data augmentation techniques.

Table 4: Ablation study on the effect of DDIM steps in the SSL inference process.

| ID | DDIM steps in inference | ScanNet 5% | |
|---|---|---|---|
| | | mAP @ 0.25 | mAP @ 0.5 |
| (1) | 0 | $42.8 \pm 0.5$ | $26.8 \pm 0.6$ |
| (2) | 1 | $43.1 \pm 0.4$ | $27.0 \pm 0.7$ |
| (3) | 2 | $\mathbf{43.5} \pm 0.2$ | $\mathbf{27.9} \pm 0.3$ |

Table 5: Ablation study on the effect of diffusion process in the different stages.

| ID | Diffusion training | DDIM in inference | ScanNet 5% | |
|---|---|---|---|---|
| | | | mAP @ 0.25 | mAP @ 0.5 |
| (1) | | | $40.5 \pm 1.2$ | $22.8 \pm 0.8$ |
| (2) | ✓ | | $42.8 \pm 0.5$ | $26.8 \pm 0.6$ |
| (3) | ✓ | ✓ | $\mathbf{43.5} \pm 0.2$ | $\mathbf{27.9} \pm 0.3$ |

Table 6: Performance of our diffusion model with different numbers of proposals and DDIM steps on the ScanNet and SUN RGB-D datasets, where the runtime in FPS and accuracy in mAP are reported.

| Dataset | Model | Num. proposal | DDIM | FPS | mAP@0.25 | mAP@0.5 |
|---|---|---|---|---|---|---|
| ScanNet (5%) | 3DIoUMatch | 128 | - | 50.75 | 39.2 | 23.1 |
| | Ours | 128 | 1 | 36.13 | 42.9 | 26.0 |
| | | | 2 | 30.07 | 43.8 | 28.0 |
| | | 256 | 1 | 28.91 | 44.6 | 27.2 |
| | | | 2 | 21.29 | 44.9 | 28.3 |
| SUN RGB-D (1%) | 3DIoUMatch | 128 | - | 65.54 | 21.3 | 8.0 |
| | Ours | 128 | 1 | 46.64 | 29.6 | 12.9 |
| | | | 2 | 36.52 | 31.3 | 14.6 |
| | | 256 | 1 | 38.10 | 30.8 | 13.6 |
| | | | 2 | 28.14 | 31.8 | 14.7 |

## B.3 Diffusion Process in Training and Inference

Our diffusion mechanism operates during both the training and inference stages of the SSL process. During SSL training, the diffusion reverse process occurs in the teacher model, denoising noisy data to produce high-quality pseudo-labels through diffusion sampling. Subsequently, the diffusion forward process takes place in the student model, learning to predict objects from the corrupted (pseudo) ground-truth. In the SSL inference stage, similar to other diffusion models, our Diffusion-SS3D is utilized to denoise randomly generated inputs. Specifically, our diffusion model is initially trained under the SSL setting. During inference, the model denoises inputs of various object sizes and label distributions through DDIM sampling, generating precise final predictions. Despite the diffusion model's primary purpose being denoising during inference, users have the flexibility to use our learned decoder and output model predictions directly, bypassing the denoising step. The results presented in Table 4 on ScanNet demonstrate our model's competitive performance even without denoising (indicated by a DDIM step of 0) due to our comprehensive diffusion model training. Additionally, employing more DDIM steps further enhances performance.

Moreover, to emphasize our diffusion model's effectiveness across different phases, we provide results for both training and inference phases on ScanNet. We compare scenarios with the diffusion process applied in both phases, training alone, and a baseline method (3DIoUMatch [4]) in Table 5. Remarkably, our diffusion model significantly improves results compared to the 3DIoUMatch [4], even when the denoising step during inference is omitted. When the denoising process is incorporated during inference, the results are further improved, underscoring the robustness of our approach.

## B.4 Number of Proposal Boxes

One limitation of the diffusion model lies in its computational costs. To address this, striking a balance between efficiency (FPS) and accuracy (mAP) is crucial. In Table 6, we report the effects of using different amounts of proposals with different numbers of DDIM steps in our diffusion model, where the proposal amount is set to 128 or 256 while using one or two DDIM steps. Compared with

the 3DIoUMatch [4] baseline, our method with DDIM sacrifices some efficiency but significantly improves mAP performance. For instance, with 128 proposals and one DDIM step on the SUN RGB-D dataset, our runtime speed decreases by 28.8% (from 65.54 FPS to 46.64 FPS), while mAP@0.5 sees a 61.3% relative improvement (from 8% to 12.9%) compared to our baseline method without diffusion, i.e., 3DIoUMatch [4]. To balance the trade-off between efficiency (FPS) and accuracy (mAP) and ensure fair comparisons with baseline methods using 128 proposals, our method utilizes 128 proposals in all experiments presented in the main paper. These results demonstrate our ability to adjust diffusion sampling steps, achieving a balance between accuracy and efficiency, effectively addressing the runtime limitation.

## B.5 Qualitative Visualization

In Figure 1, we demonstrate the effectiveness of our diffusion model by visualizing the quality of pseudo-labels on the unlabeled training set generated by the teacher model during SSL training. This visualization supports our claim that our diffusion model provides more reliable pseudo-labels, significantly enhancing the SSL training pipeline. Furthermore, in Figure 2 and Figure 3, we illustrate the effectiveness of our diffusion model in the denoising process. We first plot noisy random bounding boxes for a given point cloud scene. Then, the DDIM step is performed to obtain the denoised bounding boxes, where most noisy bounding boxes are refined toward ground truths, and those closest to the ground truth are highlighted in red. Our diffusion decoder yields the detection results with more accurate object bounding boxes and category predictions. These illustrations demonstrate the effectiveness of our diffusion model in denoising from noisy bounding boxes, leading to notable improvements in 3D object detection in a semi-supervised way.

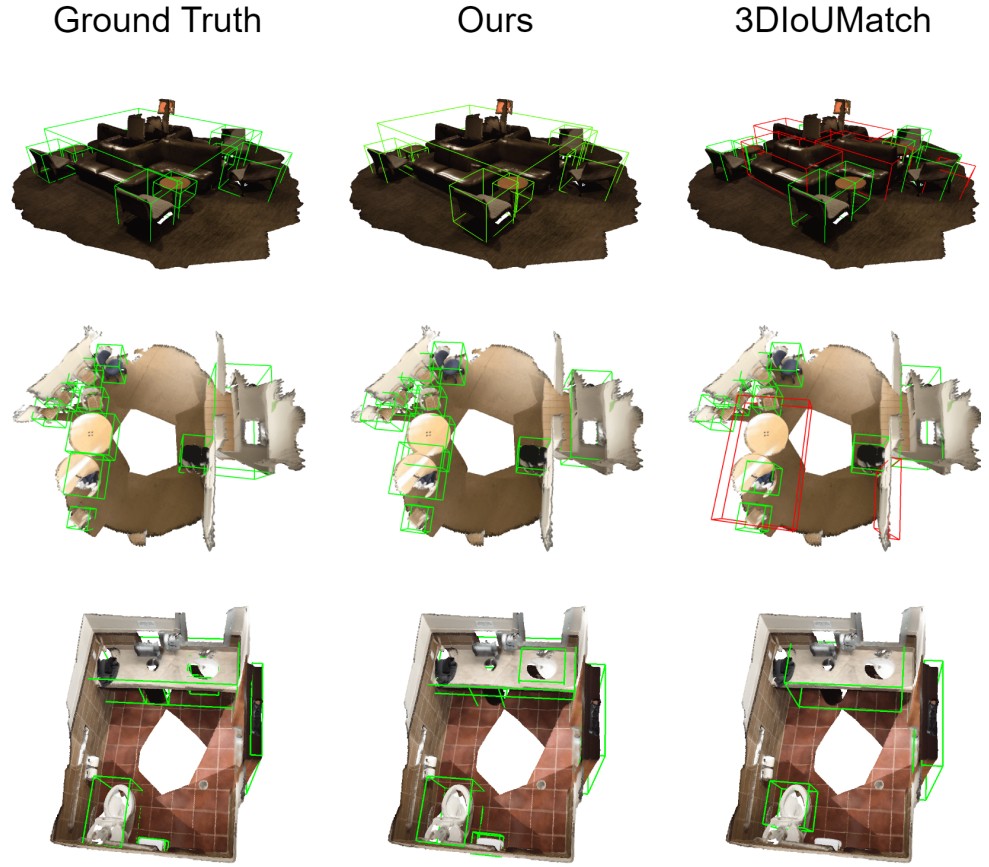

Figure 1: **Example results of generated pseudo-labels on the unlabeled training set from ScanNet.** The green bounding boxes represent proposals with an IoU score exceeding 0.25 with respect to the ground truth, while the red bounding boxes indicate false positives with an IoU score lower than 0.25. Compared to 3DIoUMatch, our method with the diffusion process is able to discover more true objects and generate less false positives as pseudo-labels.

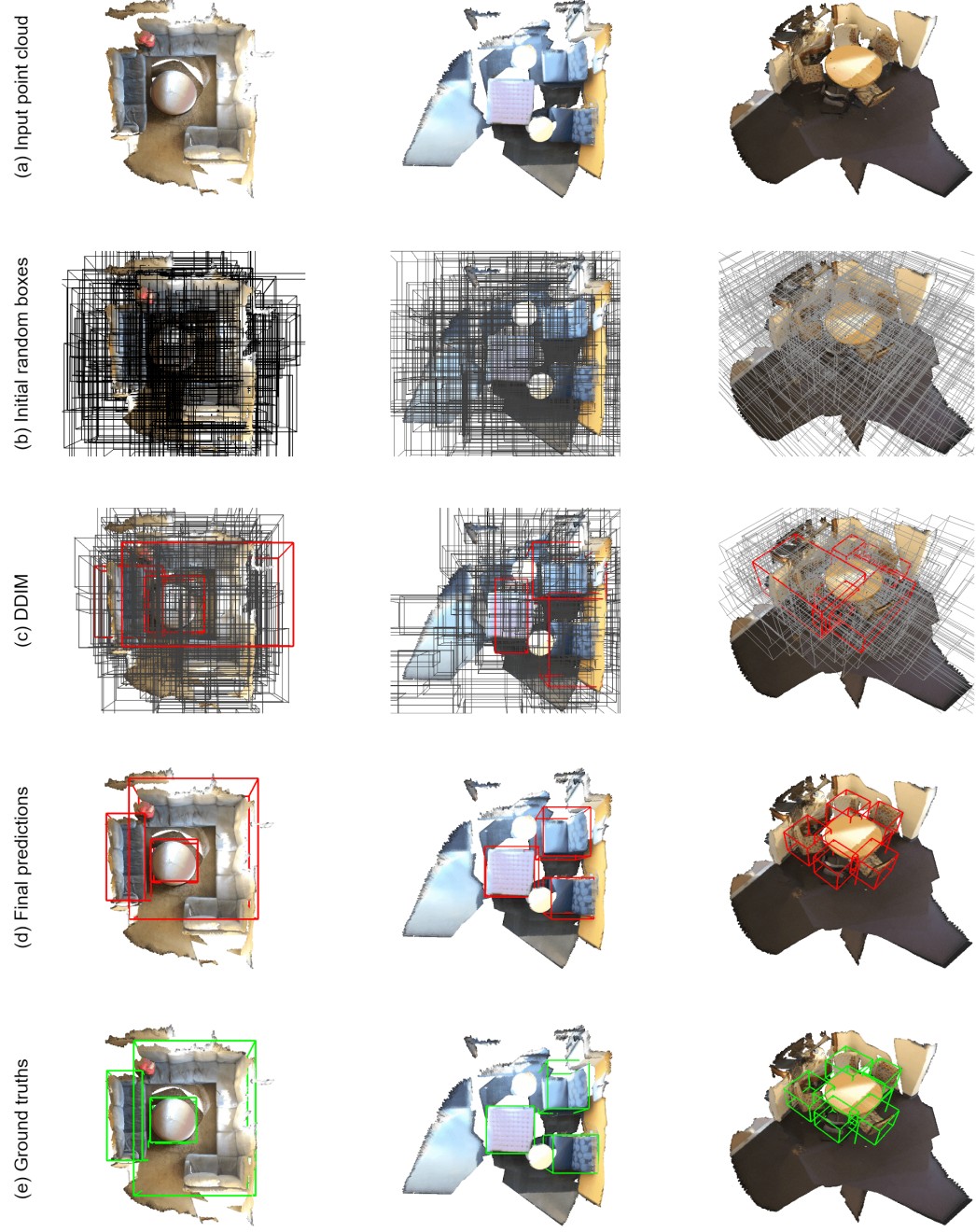

Figure 2: **Visualization of the DDIM sampling step during inference.** In each example (column), we show (a) the input point cloud, (b) the initial bounding boxes obtained by random sampling, (c) the denoised bounding boxes yielded by DDIM where those closest to the ground truth are highlighted in red, (d) the detection results given by our diffusion decoder, and (e) the ground-truth bounding boxes.

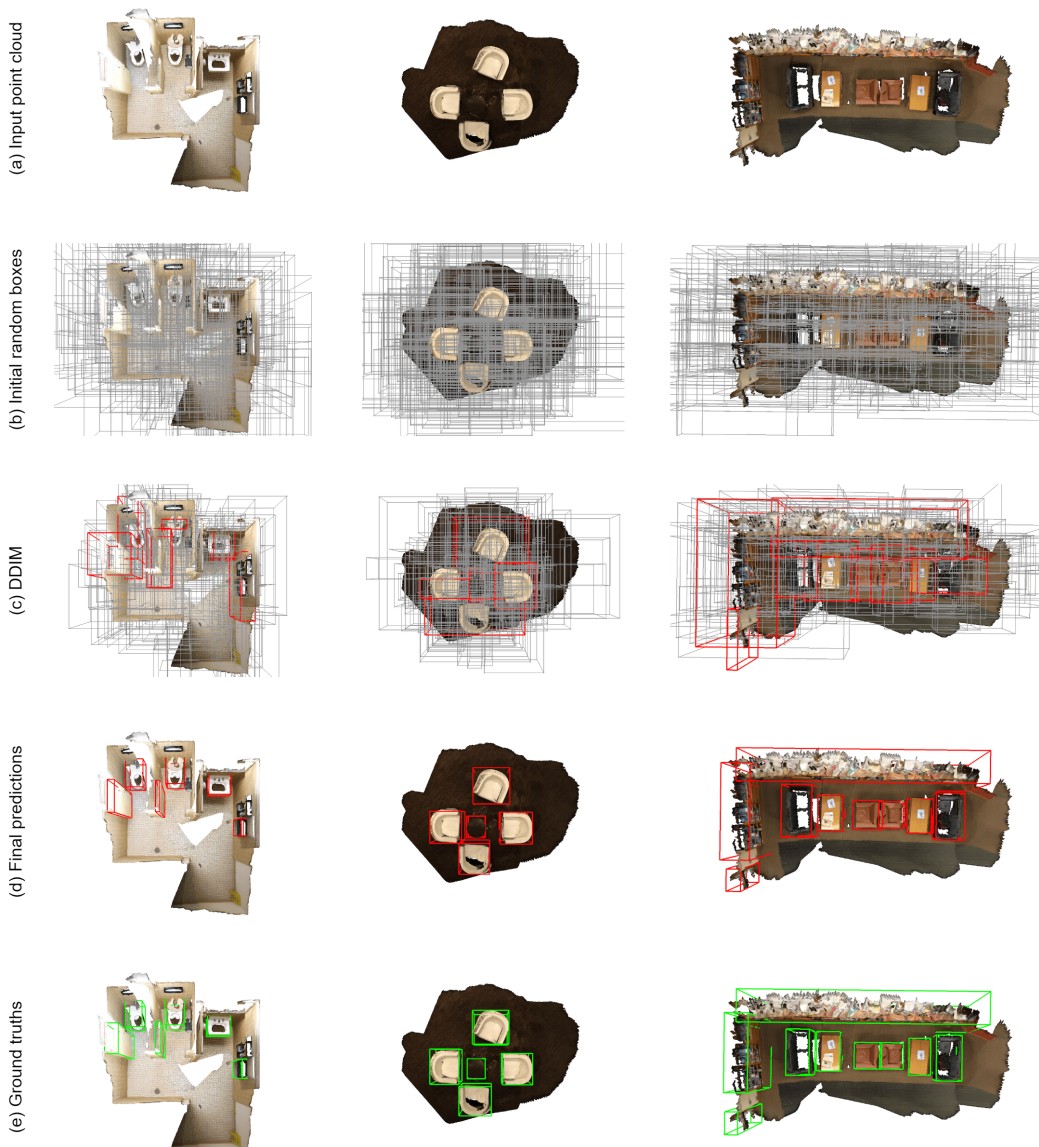

Figure 3: **Visualization of the DDIM sampling step during inference.** In each example (column), we show (a) the input point cloud, (b) the initial bounding boxes obtained by random sampling, (c) the denoised bounding boxes yielded by DDIM where those closest to the ground truth are highlighted in red, (d) the detection results given by our diffusion decoder, and (e) the ground-truth bounding boxes.