# OpenReview forum: "Diffusion-SS3D: Diffusion Model for Semi-supervised 3D Object Detection"
_NeurIPS.cc/2023/Conference — NeurIPS 2023 poster_

### Official Review · Reviewer_5wDL · 2023-06-28

**Soundness:** 3 good
**Presentation:** 3 good
**Contribution:** 3 good
**Rating:** 6
**Confidence:** 4

**Summary:**

The paper addresses the problem of semi-supervised 3D object detection (3DOD) where the aim is to train a 3DOD model using only few labelled and a lot of unlabelled data. On top of previous approaches, the authors propose to use the diffusion mechanism to enhance the pseudo labels generated by a teacher model for the unlabeled scenes. The main idea is to denoise noisy initial bounding boxes and class label distributions via the diffusion mechanism and thereby obtain better pseudo labels, which are used to train a student model. Additionally, the authors use the diffusion-based denoising to enhance the initial output of the student model during inference. Experiments on ScanNet and SUN RGB-D show that the approach outperforms previous approaches.

**Strengths:**

1. Abstract and introduction provide a clear motivation for the considered problem as well as for the proposed solution. The contributions over previous works are clearly stated and discussed.
2. The method description and the mathematical definitions are clear and easy to follow. Figures and the algorithm are nicely prepared and facilitate the method description’s clarity.
3. The proposed method clearly outperforms the considered baselines.
4. Various ablation studies show the effect of various method components and parameters.
5. Nice qualitative examples in the supplementary. It would be nice to refer to them in the main paper.

**Weaknesses:**

1. For a large part of the paper it remained a bit unclear to me, if the author’s method is applied during training or during inference. In the abstract and introduction, it first sounds like the method is applied only during training, but later on it becomes clear that the diffusion mechanism is also applied during inference to refine the initial predictions. I think this should be clearly stated in abstract and introduction.
2. In the SOTA comparison it remains unclear to me, which part of the improvement is due to the enhanced pseudo labels during training and which part is due to the refinement of predictions during inference. It would help, if the baseline without the author’s method could be added to the Table. Additionally, it would help, if a variant of the method using only training-side improvement but not the prediction enhancement during inference would be added. Thereby, it becomes clear, how much improvement can be achieved without additional inference complexity and how much improvement can be achieved due to enhancement of the predictions at the cost of additional inference complexity.

Although, there are a few weaknesses in this paper, I think this work could present a valuable contribution to NeurIPS, if the mentioned issues can be addressed in the rebuttal.

Minor comments and suggestions:

3. In Figure 1, it would help the clarity if the abbreviations such as EMA were not abbreviated.
4. Section 3.1: the symbol l is used two times with different meaning: First, it is an index for labelled (line 116) and then as an index for length (line 121).
5. The second paragraph in Section 4.1 discusses results which I could not find in any of the tables. It would help to add the comparison to, e.g., Tables 1 and 2.
6. The baselines in the ablations in Tables 3 and 4 are exactly equal to the results of 3DIoUMatch. Did the authors resimulate those results? I think it would make sense to resimulate them for the ablation studies to avoid impact of differing hardware or hyperparameters on the ablation results.

**Questions:**

1. The authors have evaluated their method on two indoor datasets. However, many recent works, e.g., [a], [b], mainly evaluate on outdoor datasets for 3DOD. Have the authors tried, how their method performs on such datasets? Maybe it makes sense to mention as a limitation that the method’s efficacy has not yet been established for such outdoor datasets.
2. The authors claim that the pseudo labels are improved by the diffusion mechanism. Is there an evaluation that supports this claim, i.e., an evaluation of the pseudo labels of the author’s method vs. a baseline without diffusion?
3. As far as I understood, the authors make use of the diffusion mechanism in two ways: First, the pseudo labels are improved and second as a refinement technique of 3DOD prediction during inference. I was wondering, if the second aspect is limited to semi-supervised 3DOD? Could it also be used to enhance fully supervised 3DOD?

[a] Park et al., “DetMatch: Two Teachers are Better than One for Joint 2D and 3D Semi-Supervised Object Detection,” ECCV 2022.

[b] Lian et al., “Semi-supervised Monocular 3D Object Detection by Multi-view Consistency,” ECCV 2022.

**Limitations:**

Limitations of the method have been discussed to some degree. I think it would be interesting to additionally state if the method has limitations w.r.t. outdoor datasets and/or in terms of added complexity during training or inference.

---

> ### Author Rebuttal · Authors · 2023-08-09
>
> Thanks for your constructive feedback and we address each question below.
>
> **Q1: Diffusion applied during training or inference.**
>
> Our diffusion mechanism is applied during both the training and inference stages. However, like most diffusion models, our Diffusion-SS3D is applied to denoise randomly generated inputs during inference. To be specific, our diffusion model is first trained under the SSL setting. Then, during inference, the model would denoise from randomly generated object sizes and label distributions via DDIM sampling to generate final predictions.
>
> Although the diffusion model is designed to perform denoising during inference, it is still possible to directly use our learned decoder and output model predictions without the denoising step. We show results in the table below on ScanNet, where the DDIM step equal to 0 indicates no denoising step during inference. Our model still performs competitively due to our diffusion model training stage. The performance is improved further using more DDIM steps. We will clarify this in the final version and add more results.
>
> DDIM steps in inference|5% mAP@0.25|5% mAP@0.5|
> |:---:|:---:|:---:|
> |0|42.8 ± 0.5|26.8 ± 0.6|
> |1|43.1 ± 0.4|27.0 ± 0.7|
> |2|43.5 ± 0.2|27.9 ± 0.3|
>
> **Q2: Improvement due to the enhanced pseudo labels during training or inference, and its complexity.**
>
> As indicated in Q1 above, we report results of having the diffusion process in both training and inference or merely training on ScanNet, as well as the baseline without our method (i.e., 3DIoUMatch). Overall, with diffusion model training but without the denoising step during inference, the results are improved significantly from the 3DIoUMatch baseline while having the same runtime speed. With the denoising process during inference, the results are improved further.
>
> |Diffusion training|DDIM in inference|5% mAP@0.25|5% mAP@0.5|
> |:---:|:---:|:---:|:---:|
> |||40.5 ± 1.2|22.8 ± 0.8|
> |✓||42.8 ± 0.5|26.8 ± 0.6|
> |✓|✓|43.5 ± 0.2|27.9 ± 0.3|
>
> Regarding the inference complexity, we show results in Table 4 of the supplementary material. For example, compared to 3DIoUMatch, our runtime speed can be decreased by 28.8% (from 65.54 FPS to 46.64 FPS) while the performance in mAP@0.5 relatively improves by 61.3% (from 8% to 12.9%) on SUN RGB-D (Ln 61-65 of the supplementary material).
>
> **Q3: Minor comments.**
>
> We will revise the paper as suggested, including EMA in Figure 1 and notations in Section 3.1.
>
> For the second paragraph of Section 4.1, we discuss the usage of applying existing augmentation methods. Due to limited space, we only present this discussion in the text and include Table 3 in the supplementary material. We will improve the presentation.
>
> For baselines in Tables 3 and 4, we re-run 3DIoUMatch with comparable implementations and hardware on ScanNet. We obtain very similar results to the original ones shown in the table below.
>
> 3DIoUMatch|5% mAP@0.25|5% mAP@0.5|
> |:---:|:---:|:---:|
> Original |40.0 ± 0.9|22.5 ± 0.5|
> Re-simulated |40.5 ± 1.2|22.8 ± 0.8|
>
> **Q4: Outdoor dataset.**
>
> Due to limited time to experiment with a new training framework (i.e., the outdoor dataset needs a different baseline than VoteNet used in the paper), we will include the results in the final version. Note that, by design of the proposed diffusion model in a general teacher-student framework, our method should not be limited to certain datasets. However, we do recognize that some changes may be needed as some object properties (e.g., location, density) in outdoor would be different from the indoor scenario. We will add this discussion to the final paper.
>
> In addition, thanks for providing the references [a, b], and we will discuss them in the final paper. Note that, [a] operates a multi-view setting using stereo or video data, while [b] considers multi-modal data with both 2D and 3D labeled data, which are different from the setting of this work.
>
> **Q5: Pseudo-label quality.**
>
> To validate whether the quality of pseudo-labels is improved, we evaluate the metrics on unlabeled training data during model training, via the teacher model that generates pseudo-labels. In the table below, overall the pseudo-label quality of our Diffusion-SS3D is better than 3DIoUMatch by more than 8% improvement in mAP and recall rate. In addition, we observe that our diffusion model achieves a better quality in earlier epochs, and then become stable during the entire training process. Note that, we still observe the improvement of SSL performance during training, since the model needs to be trained longer and learn from both labeled and unlabeled data. We will add them to the final version.
>
> | ScanNet 5% |Metric|Epoch 100|Epoch 200|Epoch 400|Epoch 800| Epoch 1000|
> |:---:|:---:|:---:|:---:|:---:|:---:|:---:|
> | 3DIoUMatch | mAP@0.5 |14.08|17.85|21.73|22.17|22.42|
> |  | Recall@0.5 |27.86|31.49|35.24|36.61|35.25|
> | Diffusion-SS3D | mAP@0.5 | 29.98 | 30.09|30.86| 31.01 |30.93 |
> |  | Recall@0.5 | 43.73 | 44.14 |45.06|44.72|44.17|
>
> **Q6: Fully-supervised setting.**
>
> Our method is indeed not limited to only the SSL setting, and we can still train the model in fully-supervised setting without using our components relevant to unlabeled data. Specifically, we train the diffusion model with 100% labeled data using the same way as the student model does (top of Figure 2 in the main paper). During inference, random data distribution is then generated (Figure 3 of the main paper) and denoised via DDIM sampling to produce final predictions.
>
> We show the result in the table below on ScanNet, where our method performs better than the baseline without diffusion by more than 1%. Although the fully-supervised setting is not our main focus, this demonstrates the potential of introducing the diffusion process in more settings.
>
> |Model|100% mAP@0.25|100% mAP@0.5|
> |:---:|:---:|:---:|
> |SESS|61.3|38.8|
> |3DIoUMatch|62.9|42.1|
> |Diffusion-SS3D|64.1|43.2|
> |Gain (mAP)|+1.2|+1.1|
>
> **Q7: Limitations.**
>
> We will add more discussions as in Q2 and Q4.

---

> > ### Comment · Reviewer_5wDL · 2023-08-16
> >
> > Thank you very much for the detailed response and the provided clarifications as well as the additional results and ablations. I have no further questions at this point.

---

> > > ### Author Response · Authors · 2023-08-16
> > > **Thank you for your comments**
> > >
> > >
> > > Thank you for the comments. We wonder whether you could consider raising the score as all issues have been addressed. We appreicate your help.

---

> > > > ### Comment · Reviewer_5wDL · 2023-08-20
> > > >
> > > > Thank you again for the provided clarifications. After also going again through other reviewers comments and taking into acount the discussions, I will raise my rating to Weak Accept.

---

> ### Author Response · Authors · 2023-08-14
> **Please let us know whether you have additional questions after reading our response**
>
> We appreciate your reviews and comments. We hope our responses address your concerns. Please let us know if you have further questions after reading our rebuttal.
>
> We hope to address all the potential issues during the discussion period.
>
> Thank you.

---

### Official Review · Reviewer_powF · 2023-07-04

**Soundness:** 2 fair
**Presentation:** 3 good
**Contribution:** 2 fair
**Rating:** 5
**Confidence:** 5

**Summary:**

This paper introduces a novel approach to enhance the accuracy of pseudo-labels and inference results in semi-supervised 3D object detection through the utilization of diffusion models. The authors propose the integration of diffusion models from two perspectives: 3D object sizes and class label distributions. The application of diffusion models to 3D object sizes aims to improve the recall rate of ground-truth objects, while the denoising process of class label distributions addresses the issue of low accuracy in category predictions for pseudo-labels. The paper includes comprehensive ablation studies and detailed analysis to validate the effectiveness of the proposed approach.

**Strengths:**

1. The authors present a comprehensive and detailed explanation of the algorithm, providing readers with a clear understanding of the proposed method and its underlying principles.
2. The use of diffusion models to improve the quality of pseudo labels in semi-supervised 3D object detection is novel.


**Weaknesses:**

1. The authors should include a comparison on the 20% labeled data, as it is a commonly used setting in the literature, such as in SESS and 3DIoUMatch. Additionally, it would be beneficial to include comparisons at larger labeled ratios, such as 50% and 100%, to provide a more comprehensive analysis of the proposed method's performance.

2. Although the use of representative points in the proposed diffusion model for 3D object sizes is mentioned as a distinction from the 2D object detection approach [9], it is a straightforward idea as point sub-sampling is commonly employed in 3D object detection to reduce the search space. Moreover, it would be valuable to investigate the effects of using different sub-sampling techniques, such as random sub-sampling, and compare them with FPS sampling.

3. The diffusion model for refining bounding boxes appears similar to the IoU Optimization technique proposed in 3DIoUMatch, which aims to optimize the center and size of bounding boxes. It would be informative to clarify whether the IoU Optimization technique has been applied to the pseudo labels when comparing them with the results from 3DIoUMatch.


**Questions:**

Is there any existing research that demonstrates the effectiveness of denoising classification logits through the diffusion process on label distributions in other classification-based tasks, such as image classification? It would be beneficial to discuss in the related work section if there are any existing studies.

**Limitations:**

The authors did not discuss the limitations or potential negative social impact of the proposed method.

---

> ### Author Rebuttal · Authors · 2023-08-09
>
> Thanks for your constructive feedback and we address each question below.
>
> **Q1: Comparisons on the 20% labeled data or larger labeled ratios.**
>
> For ScanNet, the results of using 20% labeled data are reported in Table 1 of the main paper. For SUN RGB-D, we show in the table below that our Diffusion-SS3D performs better than 3DIoUMatch by more than 1.5% in both mAP@0.25 and mAP@0.5. Due to limited time and computational resources, we present the new results of using one random data split and will include full results using more random splits in the final version.
>
> | SUN RGB-D | 20% mAP@0.25 | 20% mAP@0.5 |
> |:---:|:---:|:---:|
> | VoteNet | 45.7 ± 0.6 | 22.5 ± 0.8 |
> | SESS | 47.1 ± 0.7 | 24.5 ± 1.2 |
> | 3DIoUMatch | 49.7 ± 0.4 | 30.9 ± 0.2 |
> | Diffusion-SS3D | **51.3** | **32.7** |
> | Gain (mAP) | +1.6 | +1.8 |
>
> For higher ratios of labeled data or the fully-supervised setting (i.e., 100%), we did run the 100% setting on ScanNet and the results are shown in the table below. We find that our Diffusion-SS3D has more than 1% improvement in both mAP metrics. In addition, we also include the results of 20% labeled data as reference (also shown in Table 1 of the main paper). Note that, although the fully-supervised setting is not our main focus in this paper, these results demonstrate the potential of introducing the diffusion process in more settings.
>
> | ScanNet | 20% mAP@0.25 | 20% mAP@0.5 | 100% mAP@0.25 | 100% mAP@0.5 |
> |:---:|:---:|:---:|:---:|:---:|
> | VoteNet | 46.9 ± 1.9 | 27.5 ± 1.2 | 57.8 | 36.0 |
> | SESS | 49.6 ± 1.1 | 29.0 ± 1.0 | 61.3 | 38.8 |
> | 3DIoUMatch | 52.8 ± 1.2 | 35.2 ± 1.1 | 62.9 | 42.1 |
> | Diffusion-SS3D | **55.6** ± 1.7 | **36.9** ± 1.4 | **64.1** | **43.2** |
> | Gain (mAP) | +2.8 | +1.7 | +1.2 | +1.1 |
>
> **Q2: Effects of using different sub-sampling techniques.**
>
> Although we are aware of point sub-sampling as a common strategy in 3D object detection, we find that leveraging this scheme in our diffusion model can reduce the search space and decrease the degree of freedom in generated noisy boxes, facilitating the diffusion learning process, in which the 2D case [9] does not have the similar issue.
>
> We show more results in the table below regarding the point sampling strategies. With the suggested random sub-sampling, we find that results are similar to the ones using FPS (i.e., within 0.5% differences in mAP), for both models with and without our diffusion component. In practice, we follow the implementation of the VoteNet and 3DIoUMatch methods that use FPS. We will add these results in the final version.
>
> | ID | Diffusion | Sampling | ScanNet 5% mAP@0.25 | ScanNet 5% mAP@0.5 |
> |:---:|:---:|:---:|:---:|:---:|
> | (1) |  | Random | 40.2 ± 1.5 | 22.1 ± 1.1 |
> | (2) |  | FPS | 40.0 ± 0.9 | 22.5 ± 0.5 |
> | (3) | ✓ | Random | 43.1 ± 0.6 | 27.4 ± 0.6 |
> | (4) | ✓ | FPS | 43.5 ± 0.2 | 27.9 ± 0.3 |
>
> **Q3: IoU optimization technique from 3DIoUMatch.**
>
> We first note that the 3D IoU prediction in 3DIoUMatch is used as a filtering scheme based on model outputs to find high-quality pseudo-labels. In contrast, our diffusion model learns to denoise from random box sizes and labels to form pseudo-labels. Therefore, applying 3DIoUMatch's filtering scheme complements our method as a post-processing step. In practice, since we consider 3DIoUMatch as our baseline, we do use the same filtering step in our framework after the diffusion steps. We will clarify it in the final version.
>
> **Q4: Existing works using diffusion process on label distributions.**
>
> Thanks for the suggestion. Since using diffusion models for recognition tasks are relatively new, we have tried our best to include the work we are aware of in the Related Work section of the main paper. We do find two concurrent works [A, B] (both published on arXiv after the NeurIPS submission deadline) that involve the diffusion process for image classification. [A] uses the embedding of the image generation model for learning a classifier, while [B] tackles the learning process of noisy labeled data via the diffusion model. We will include and discuss both works in the final version.
>
> [A] Mukhopadhyay et al., Diffusion Models Beat GANs on Image Classification, arXiv:2307.08702, July 2023.
>
> [B] Chen et al., Label-Retrieval-Augmented Diffusion Models for Learning from Noisy Labels, arXiv:2305.19518, May 2023.
>
> **Q5: Limitations and social impact.**
>
> Since the diffusion model requires more computational powers for training and inference (runtime in frame-per-second is presented in Table 4 of the supplementary material), optimizing efficiency is crucial for real-time applications and large-scale deployments. In the meantime, the increased energy consumption may cause an environmental impact, so it is worth exploring more eco-friendly computing strategies to reduce the environmental footprint. We will include more discussions in the final version.

---

> ### Author Response · Authors · 2023-08-14
> **Please let us know whether you have additional questions after reading our response**
>
> We appreciate your reviews and comments. We hope our responses address your concerns. Please let us know if you have further questions after reading our rebuttal.
>
> We hope to address all the potential issues during the discussion period.
>
> Thank you.

---

> > ### Comment · Reviewer_powF · 2023-08-17
> > **Post-rebuttal Comments**
> >
> > Thank you for addressing my concerns and providing additional experiments in your responses. Most of my concerns have been resolved, thus I am inclined to raise my rating to borderline accept. However, I would like to suggest that you include more results, such as multiple runs of the experiments, results on different labeled data ratios (e.g., 50%) and other datasets (e.g., SUN RGB-D and outdoor KITTI dataset as pointed by Reviewer TXhm), in the final version of the paper. Additionally, it would be beneficial to see the promised modifications incorporated as well.

---

> > > ### Author Response · Authors · 2023-08-17
> > > **Thank you**
> > >
> > > Dear Reviewer,
> > >
> > > We appreciate your comments and help. We will incorporate more results and revise this paper based on your comments.
> > >
> > > Thank you!

---

### Official Review · Reviewer_PTFH · 2023-07-06

**Soundness:** 2 fair
**Presentation:** 2 fair
**Contribution:** 2 fair
**Rating:** 5
**Confidence:** 4

**Summary:**

In this paper, the author argues that previous 3D semi-supervised detection methods relied solely on teacher models, which cannot generate sufficiently reliable pseudo-labels. Therefore, the author proposes a method called Diffusion-SS3D. This method enables diffusion learning to remove noise from corrupted 3D object size and class label distributions, thereby optimizing the pseudo-labels generated by the teacher model and obtaining more reliable pseudo-labels. The proposed method achieves state-of-the-art performance on the ScanNet and SUN RGB-D datasets.


**Strengths:**

Compared to previous works such as 3DIoUMatch, the approach of utilizing diffusion for denoising instead of relying solely on threshold filtering to generate better pseudo-labels is intriguing.


**Weaknesses:**

1.In my opinion, the paper suggests that diffusion can generate more reliable pseudo-labels, but it does not clearly explain why the pseudo-labels generated by diffusion are more reliable than those generated by previous works.

2.In Table 4, the performance of using only Box Renewal is not provided. I believe that both DDIM and Box Renewal have certain denoising capabilities. If the paper considers DDIM to be effective, it should provide the results of using only Box Renewal to further demonstrate this viewpoint.

**Questions:**

1.I am curious to know how much difference there would be in the performance if we trained diffusion and generated pseudo-labels using a sampled object size and class distribution based on the prior knowledge obtained from the dataset labels.

2.Since diffusion is just one module within the entire teacher-student framework, the metrics on the dataset may not directly indicate the reliability of diffusion as a source of pseudo-labels. Could the authors design an experiment that demonstrates the improved quality of pseudo-labels obtained through diffusion denoising?


**Limitations:**

1.The authors did not provide a detailed discussion on the limitations of the method. They only briefly mentioned in the final part of the experiment section that diffusion does not perform well in denoising orientations. In this case, could further exploration be conducted to investigate the conditions under which the proposed method performs well in terms of ground truth and predictions?

2.From my perspective, the diffusion model has certain computational costs, and it may be worth discussing the practicality of the method
from this angle.

---

> ### Author Rebuttal · Authors · 2023-08-09
>
> Thanks for your constructive feedback and we address each question below.
>
> **Q1: Why are the pseudo-labels generated by diffusion are more reliable?**
>
> In Figure 1 of the main paper, we illustrate the fundamental difference between the conventional framework and our diffusion model in pseudo-label generation, which results in different pseudo-label qualities. First, prior works like 3DIoUMatch are designed to refine pseudo-labels purely based on model outputs, in which the objects may not be discovered if the model cannot output sufficient predictions (i.e., lower recall rate).
>
> On the other hand, our diffusion process starts with random box sizes and label distributions, which do not depend on model predictions, so there is a higher chance (i.e., higher recall rate) we can find more objects (please refer to the table provided in Q4). Then, through the denoising process in our diffusion model, pseudo-labels are refined to be more reliable. We will add this discussion to the paper.
>
>
> **Q2: Performance of box renewal.**
>
> We experiment with only using box renewal in our diffusion framework, as shown in the table below. We observe that both DDIM and box renewal have the denoising capability. Note that the effectiveness of box renewal also benefits from the diffusion training process, in which box renewal requires the trained diffusion decoder (as shown in Algorithm 1 of the main paper), so that the decoder can take the updated bounding box features and label distributions from box renewal. Therefore, we would emphasize that the entire diffusion training framework makes major improvements rather than a single DDIM or box renewal component, i.e., box renewal may not be solely a method without the diffusion training step. We will add this result with more discussions to the final version.
>
> | ID | DDIM | Box Renewal | ScanNet 5% mAP@0.25 | ScanNet 5% mAP@0.5 |
> |:---:|:---:|:---:|:---:|:---:|
> | (1) |  |  | 40.5 ± 1.2 | 22.8 ± 0.8 |
> | (2) | ✓ |  | 42.8 ± 0.5 | 26.6 ± 0.9 |
> | (3) |  | ✓ | 42.3 ± 0.4 | 26.8 ± 0.3 |
> | (4) | ✓ | ✓ | **43.5** ± 0.2 | **27.9** ± 0.3 |
>
> **Q3: Prior knowledge in diffusion training.**
>
> Thanks for the suggestion. Using prior knowledge in the diffusion process is interesting, but this may require more studies. For instance, we may consider using category-specific random sampling based on the possible object size as the prior. However, this may be non-trivial as we do not assume to know which category to perform random sampling during inference. In practice, we experiment and find that randomly sampling the object size within 1/4 of the entire scene is more effective than using a larger object size. This supports that having some prior knowledge should be still a helpful cue and we will consider this as a future work.
>
> **Q4: Pseudo-label quality.**
>
> To validate whether the quality of pseudo-labels is improved, we evaluate the metrics on unlabeled training data during model training, via the teacher model that generates pseudo-labels. In the table below, we show that overall the pseudo-label quality of our Diffusion-SS3D is better than 3DIoUMatch by more than 8% improvement in mAP and recall rate. In addition, we observe that our diffusion model is able to achieve a better quality in earlier epochs, and then become stable during the entire training process. Note that, we still observe the improvement of semi-supervised performance during training, since the model needs to be trained longer and learn from both labeled and unlabeled data. These results support our claim that the diffusion model can generate high-quality pseudo-labels and we will add the results to the final version. In addition, we have included more visual comparisons of generated pseudo-labels in the rebuttal pdf file.
>
> | ScanNet 5% | Metric | Epoch 100 | Epoch 200 | Epoch 400 | Epoch 600 | Epoch 800 | Epoch 1000 |
> |:---:|:---:|:---:|:---:|:---:|:---:|:---:|:---:|
> | 3DIoUMatch | mAP@0.5 | 14.08 | 17.85 | 21.73 | 21.95 | 22.17 | 22.42 |
> |  | Recall@0.5 | 27.86 | 31.49 | 35.24 | 36.22 | 36.61 | 35.25 |
> | Diffusion-SS3D | mAP@0.5 | 29.98 | 30.09 | 30.86 | 30.55 | 31.01 | 30.93 |
> |  | Recall@0.5 | 43.73 | 44.14 | 45.06 | 44.75 | 44.72 | 44.17 |
>
> **Q5: Limitations.**
>
> In diffusion model training, it relies on ground-truth information to perform the denoising process. For the datasets we experiment on, we find that the orientation information may not be available or inaccurate. For example, ScanNet does not provide orientations, i.e., all objects being asigned with orientations as 0. For SUN RGB-D, orientations are provided inconsistently across different scenes, which causes the training difficulty in denoising orientations (Ln 308-310). We will consider this as a future work to explore noisy data for diffusion models.
>
> For other limitations, since the diffusion model requires more computational powers for training and inference (runtime in frame-per-second is presented in Table 4 of the supplementary material), optimizing efficiency is crucial for real-time applications and large-scale deployments. In the meantime, the increased energy consumption may cause an environmental impact, so it is worth exploring more eco-friendly computing strategies to reduce the environmental footprint. We will include more discussions in the final version.
>
> **Q6: Computational costs.**
>
> One of the limitations of the diffusion model is its computational costs as described above (highlighted in Table 4 of the supplementary material). For example, compared to our baseline method without diffusion, i.e., 3DIoUMatch, our runtime speed is decreased by 28.8% (from 65.54 FPS to 46.64 FPS) while the performance in mAP@0.5 relatively improves by 61.3% (from 8% to 12.9%) on SUN RGB-D (Ln 61-65 of the supplementary material). We will add more discussions regarding this limitation, e.g., adjusting the diffusion sampling steps to achieve a trade-off between accuracy and efficiency as shown in Table 4 of the supplementary material.

---

> ### Author Response · Authors · 2023-08-14
> **Please let us know whether you have additional questions after reading our response**
>
> We appreciate your reviews and comments. We hope our responses address your concerns. Please let us know if you have further questions after reading our rebuttal.
>
> We hope to address all the potential issues during the discussion period.
>
> Thank you.

---

> > ### Comment · Reviewer_PTFH · 2023-08-16
> >
> > Thanks for the rebuttal. With these additional results and explanations, my concerns have been addressed. I hope your final version can contribute to the development of the field.

---

> > > ### Author Response · Authors · 2023-08-16
> > > **Thank you for the comments**
> > >
> > > We thank you for the comments and will include all the discussions in the revised manuscript and supplementary material.
> > >
> > > As all the issues have been addressed, we wonder whether you could consider raising the scores. Your help is gratefully appreciated.

---

> > > > ### Comment · Reviewer_PTFH · 2023-08-17
> > > >
> > > > I have raised my rating from "boaderline reject" to "boaderline accept".

---

### Official Review · Reviewer_TXhm · 2023-07-06

**Soundness:** 4 excellent
**Presentation:** 3 good
**Contribution:** 3 good
**Rating:** 7
**Confidence:** 4

**Summary:**

This paper proposes a semi-supervied 3D object detection framework, named Diffusion-SS3D. Diffusion-SS3D introduces diffusion process to improve the quality of pseudo-labels. The authors perform experiments on ScanNet and SUN RGB-D benchmark to verify the effectiveness.

**Strengths:**

+ The motivation of this paper makes sense.
+ The experimental results are promising.
+ The combination of semi-supervised 3D object detection and diffusion is interesting and novel.

**Weaknesses:**

- The statements in the method part are too long. The writing should be organized again. Try to use more detailed figure to introduce your method.
- How to evaluate the quality of pseudo-labels? It’s better to provide statistical data or visualization to support your claims.
- In Tables 5 and 6, it seems that the diffusion steps and scaling factors for SNR are sensitive to different datasets. What is the reason for the above? Please provide some insights or analysis on this phenomenon (maybe the amount of labeled data is limited?).

**Questions:**

Please see the weaknesses.

**Limitations:**

The method proposed by the author has achieved good results in indoor datasets, e.g., ScanNet and SUN RGB-D. What about the performance on outdoor datesets? And what about the performance of the proposed method in multi-camera 3D detection methods?

---

> ### Author Rebuttal · Authors · 2023-08-09
>
> Thanks for your constructive feedback and we address each question below.
>
> **Q1: Pseudo-label quality.**
>
> To validate whether the quality of pseudo-labels is improved, we evaluate the metrics on unlabeled training data during model training, via the teacher model that generates pseudo-labels. In the table below, we show that overall the pseudo-label quality of our Diffusion-SS3D is better than 3DIoUMatch by more than 8% improvement in mAP and recall rate. In addition, we observe that our diffusion model is able to achieve a better quality in earlier epochs, and then become stable during the entire training process. Note that, we still observe the improvement of semi-supervised performance during training, since the model needs to be trained longer and learn from both labeled and unlabeled data. These results support our claim that the diffusion model can generate high-quality pseudo-labels and we will add them to the final version. In addition, we have included more visual comparisons of generated pseudo-labels in the rebuttal pdf file.
>
> | ScanNet 5% | Metric | Epoch 100 | Epoch 200 | Epoch 400 | Epoch 600 | Epoch 800 | Epoch 1000 |
> |:---:|:---:|:---:|:---:|:---:|:---:|:---:|:---:|
> | 3DIoUMatch | mAP@0.5 | 14.08 | 17.85 | 21.73 | 21.95 | 22.17 | 22.42 |
> |  | Recall@0.5 | 27.86 | 31.49 | 35.24 | 36.22 | 36.61 | 35.25 |
> | Diffusion-SS3D | mAP@0.5 | 29.98 | 30.09 | 30.86 | 30.55 | 31.01 | 30.93 |
> |  | Recall@0.5 | 43.73 | 44.14 | 45.06 | 44.75 | 44.72 | 44.17 |
>
> **Q2: Sensitivity in Tables 5 and 6.**
>
> As discussed in Ln 291-297 of the main paper, one reason for the sensitivity of diffusion sampling steps (Table 5 of the main paper) to different datasets is indeed the limited labeled data as the reviewer suggested. Since the diffusion training process is mainly guided by the limited labeled data, it makes diffusion sampling steps more sensitive when inferring on unlabeled data distribution. Similarly for Table 6 and Ln 298-304 of the main paper, since the scaling factor determines the difficulty of noisy data (i.e., how difficult to denoise), having limited labeled data would also make the training process more challenging. We will clarify this in the final version.
>
> Nevertheless, in both Table 5 and 6 of the main paper, no matter whether there are slight deviations across different settings, we can conclude that using DDIM step equal to 2 and the scaling factor equal to 4.0 should be a general rule-of-thumb that achieves better results compared to the baseline without diffusion.
>
> **Q3: Outdoor dataset.**
>
> Due to limited time to experiment with a new training framework (i.e., the outdoor dataset needs a different baseline than VoteNet we use in the paper), we will include the results in the final version. Note that, by design of the proposed diffusion model in a general teacher-student framework, our method should not be limited to certain datasets. However, we do recognize that some changes may be required as some object properties (e.g., location, density) in outdoor would be different from the indoor scenario.
>
> For the multi-camera setting, it is an interesting direction as it may provide more information, e.g., pseudo-label consistency from multi-views in the diffusion training process, which can be considered as a future work as it is beyond the scope of this work. We will include these discussions in the final version.
>
> **Q4: Writings.**
>
> Thank you for the feedback. We will improve the writings and figures in the final version.

---

> > ### Comment · Reviewer_TXhm · 2023-08-20
> > **Official Comments by Reviewer TXhm**
> >
> > Thanks for the rebuttal and response to my questions. Most of my concerns have been addressed, especially the “diffusion steps and scaling factors”. I keep my rating 7. Note that the writings and figures must revise in the final version.

---

> > > ### Author Response · Authors · 2023-08-21
> > > **Thank you**
> > >
> > > Dear Reviewer,
> > >
> > > We appreciate your feedback and will revise the paper based on your comments.
> > >
> > > Thank you!

---

> ### Author Response · Authors · 2023-08-14
> **Please let us know whether you have additional questions after reading our response**
>
> We appreciate your reviews and comments. We hope our responses address your concerns. Please let us know if you have further questions after reading our rebuttal.
>
> We hope to address all the potential issues during the discussion period.
>
> Thank you.

---

### Official Review · Reviewer_7AAY · 2023-07-07

**Soundness:** 3 good
**Presentation:** 3 good
**Contribution:** 3 good
**Rating:** 6
**Confidence:** 4

**Summary:**

The paper proposes a novel algorithm utilizing the diffusion model in 3D object detection for generating pseudo-labels for semi-supervised learning. Technically, it adopts the previous method of teacher-student architecture, but extends it by introdusing diffusion to generate pseudo-labels and denoising technique for class labels and object size distribution. The model incorporates both labeled and unlabeled data through asymetric augmentation mechanism.

**Strengths:**

- The novel algorithm creates high-quality pseudo-labels for unlabled data and higly increases the performance of the current techniques in 3D object detection.
- The qualitative and quantitative results of the proposed approach demonstrate good improvements over existing methods.  This indicates the effectiveness and superiority of the proposed approach in achieving better performance and accuracy for 3D object detection.
- Also, the source code is available.


**Weaknesses:**

- There needs to be ablative study for farthest point sampling. I am curious this provides the confidential results. I encourage authors to visualize its results.

- It would be helpful if there's additional ablative study for the baseline that does not use any teacher network but train the student network using the proposed method. I am curious why teacher model is necessary.

- Less qualitative results. I encourage authors to add more qualitative results.

**Questions:**

Please discuss the role of the teacher network.

**Limitations:**

I also encourage authors to add societal impact of their work.

---

> ### Author Rebuttal · Authors · 2023-08-09
>
> Thanks for your constructive feedback and we address each question below.
>
> **Q1: Farthest point sampling.**
>
> We first note that we follow the common implementation in VoteNet and 3DIoUMatch to apply farthest point sampling (FPS) in our framework. To study the impact of FPS, we further conduct an experiment to use the random point sub-sampling method. Results shown in the table below indicate that both FPS and random sampling provide competitive performance improvement compared to the 3DIoUMatch baseline. We will add this result to the final version.
>
> | ID | Diffusion | Sampling | ScanNet 5% mAP@0.25 | ScanNet 5% mAP@0.5 |
> |:---:|:---:|:---:|:---:|:---:|
> | (1) |  | Random | 40.2 ± 1.5 | 22.1 ± 1.1 |
> | (2) |  | FPS | 40.0 ± 0.9 | 22.5 ± 0.5 |
> | (3) | ✓ | Random | 43.1 ± 0.6 | 27.4 ± 0.6 |
> | (4) | ✓ | FPS | 43.5 ± 0.2 | 27.9 ± 0.3 |
>
> To visualize the point sampling results via FPS, Figure 3 of the main paper illustrates one example. The red points in the top of figure are sampled points, in which we consider them as potential object centers to further produce random object bounding boxes in the diffusion process. We will clarify this in the final version.
>
> **Q2: Why teacher model is necessary.**
>
> For semi-supervised setting, the teacher-student framework plays a key role to handle unlabeled data, where the teacher model is utilized to generate pseudo-labels from unlabeled data, serving as a supervisory signal for the learning process in the student model. In our approach, we utilize the common teacher-student framework and integrate the diffusion process, so that the diffusion sampling occuring in the teacher model generates pseudo-labels. While it may be possible to only use a single student model and generate pseudo-labels, it would not be an effective way in semi-supervised learning (e.g., the VoteNet baseline) due to the absence of a more stable teacher model (i.e., updated via an exponential moving average scheme from the student model) for pseudo-label generation. Note that, all the main methods in the paper (e.g., SESS, 3DIoUMatch) use the same teacher-student framework for fair comparisons.
>
> To realize the suggested experiment without the teacher model, we conduct an experiment of using 100% labeled data via the proposed diffusion model. In this way, there is no need to generate pseudo-labels from unlabeled data, so that the teacher model is not critical anymore. Specifically, we train our diffusion model with labeled data using the same way as the student model does (top of Figure 2 in the main paper). During inference, random data distribution is then generated (see Figure 3 of the main paper) and denoised via DDIM sampling to produce final predictions. We show the results in the table below on ScanNet, where our method performs better than the baseline without diffusion by more than 1%. Although the fully-supervised setting is not our main focus in this paper, this demonstrates the potential of introducing the diffusion process in more settings.
>
> | Model | 100% mAP@0.25 | 100% mAP@0.5 |
> |:---:|:---:|:---:|
> | VoteNet | 57.8 | 36.0 |
> | SESS | 61.3 | 38.8 |
> | 3DIoUMatch | 62.9 | 42.1 |
> | Diffusion-SS3D | **64.1** | **43.2** |
> | Gain (mAP) | +1.2 | +1.1 |
>
> **Q3: Qualitative results.**
>
> In our supplementary materials, Figure 1 and 2 demonstrate the effectiveness of our diffusion process, showcasing the progressive improvement from random noisy boxes to final predictions. In addition, we have included more visual comparisons of generated pseudo-labels in the rebuttal pdf file. We will show more qualitative results in the final version as suggested.
>
> **Q4: Societal impact.**
>
> Since the diffusion model requires more computational powers for training and inference (runtime in frame-per-second is presented in Table 4 of the supplementary material), the increased energy consumption may cause an environmental impact. Therefore, it is worth exploring more eco-friendly computing strategies to reduce the environmental footprint. We will add more discussions to the final version.

---

> ### Author Response · Authors · 2023-08-14
> **Please let us know whether you have additional questions after reading our response**
>
> We appreciate your reviews and comments. We hope our responses address your concerns. Please let us know if you have further questions after reading our rebuttal.
>
> We hope to address all the potential issues during the discussion period.
>
> Thank you.

---

### Author Rebuttal · Authors · 2023-08-10

Thanks for your constructive feedback.

In addition to addressing individual questions in each rebuttal, we include more example results of generated pseudo-labels in the pdf file, in comparisons with the 3DIoUMatch baseline that does not use the diffusion model like our method.

If there are any further inquiries, please notify us by the end of Author-Reviewer Discussion Stage (Aug 16th).

---

### Decision · Program_Chairs · 2023-09-21

**Decision:**

Accept (poster)

**Comment:**

All five reviewers recommend acceptance. The authors did a good job with their rebuttal, the reviewers concur that most concerns were addressed. For the benefit of the readers, please include the important points from the rebuttal in the final paper.